# Phosphoproteomics reveals that Parkinson's disease kinase LRRK2 regulates a subset of Rab GTPases

Martin Steger[1], Francesca Tonelli[2], Genta Ito[2], Paul Davies[2], Matthias Trost[2], Melanie Vetter[3], Stefanie Wachter[3], Esben Lorentzen[3], Graham Duddy[4†], Stephen Wilson[5], Marco AS Baptista[6], Brian K Fiske[6], Matthew J Fell[7], John A Morrow[8], Alastair D Reith[9], Dario R Alessi[2*], Matthias Mann[1*]

[1]Department of Proteomics and Signal Transduction, Max Planck Institute of Biochemistry, Martinsried, Germany; [2]Medical Research Council Protein Phosphorylation and Ubiquitylation Unit, College of Life Sciences, University of Dundee, Dundee, United Kingdom; [3]Department of Structural Cell Biology, Max Planck Institute of Biochemistry, Martinsried, Germany; [4]Molecular Discovery Research, GlaxoSmithKline Pharmaceuticals R&D, Harlow, United Kingdom; [5]RD Platform Technology and Science, GlaxoSmithKline Pharmaceuticals R&D, Stevenage, United Kingdom; [6]The Michael J. Fox Foundation for Parkinson's Research, New York, United States; [7]Early Discovery Neuroscience, Merck Research Laboratories, Boston, United States; [8]Neuroscience, Merck Research Laboratories, Westpoint, United States; [9]Neurodegeneration Discovery Performance Unit, GlaxoSmithKline Pharmaceuticals R&D, Stevenage, United Kingdom

*For correspondence: d.r.alessi@dundee.ac.uk (DRA); mmann@biochem.mpg.de (MM)

Present address: †The Wellcome Trust Sanger Institute, Hinxton, United Kingdom

**Abstract** Mutations in Park8, encoding for the multidomain Leucine-rich repeat kinase 2 (LRRK2) protein, comprise the predominant genetic cause of Parkinson's disease (PD). G2019S, the most common amino acid substitution activates the kinase two- to threefold. This has motivated the development of LRRK2 kinase inhibitors; however, poor consensus on physiological LRRK2 substrates has hampered clinical development of such therapeutics. We employ a combination of phosphoproteomics, genetics, and pharmacology to unambiguously identify a subset of Rab GTPases as key LRRK2 substrates. LRRK2 directly phosphorylates these both in vivo and in vitro on an evolutionary conserved residue in the switch II domain. Pathogenic LRRK2 variants mapping to different functional domains increase phosphorylation of Rabs and this strongly decreases their affinity to regulatory proteins including Rab GDP dissociation inhibitors (GDIs). Our findings uncover a key class of bona-fide LRRK2 substrates and a novel regulatory mechanism of Rabs that connects them to PD.

## Introduction

Parkinson's disease (PD) is the second most common neurodegenerative disease, affecting 1–2% of the elderly population (*Lees et al., 2009*). Environmental and genetic factors contribute to the development of the disease, but its precise etiology still remains elusive (*Burbulla and Krüger, 2011*). Genome-wide association studies (GWAS) have related 28 genetic risk variants at 24 loci with nonfamilial PD (*Nalls et al., 2014*). Among those, mutations in LRRK2 (Park8) are also found in hereditary forms, pinpointing a shared molecular pathway driving pathogenesis in both familial and non-familial PD and comprising the most common cause of the disease (*Simón-Sánchez et al.,*

**eLife digest** Parkinson's disease is a degenerative disorder of the nervous system that affects approximately 1% of the elderly population. Mutations in the gene that encodes an enzyme known as LRRK2 are the most common causes of the inherited form of the disease. Such mutations generally increase the activity of LRRK2 and so drug companies have developed drugs that inhibit LRRK2 to prevent or delay the progression of Parkinson's disease. However, it was not known what role LRRK2 plays in cells, and why its over-activation is harmful.

Steger et al. used a 'proteomics' approach to find other proteins that are regulated by LRRK2. The experiments tested a set of newly developed LRRK2 inhibitors in cells and brain tissue from mice. The mice had mutations in the gene encoding LRRK2 that are often found in human patients with Parkinson's disease. The experiments show that LRRK2 targets some proteins belonging to the Rab GTPase family, which are involved in transporting molecules and other 'cargoes' around cells. Several Rab GTPases are less active in the mutant mice, which interferes with the ability of these proteins to correctly direct the movement of cargo around the cell.

Steger et al.'s findings will help to advance the development of new therapies for Parkinson's disease. The next challenges are to identify how altering the activity of Rab GTPases leads to degeneration of the nervous system and how LRRK2 inhibitors may slow down these processes.

*2009*; *Satake et al., 2009*). LRRK2 encodes a large protein composed of central kinase and GTPase (ROC-COR) domains that are surrounded by multiple protein-protein interaction regions. PD pathogenic LRRK2 mutations map predominantly to the kinase (G2019S, I2020T) and the ROC-COR domains (R1441C/G/H, Y1699C), implying that these enzymatic activities are crucial for pathogenesis (*Rudenko and Cookson, 2014*). Presently, it is unclear how LRRK2 mutations occurring in different functional domains all predispose to PD. The most common PD-associated LRRK2 mutation is the G2019S amino acid substitution, which activates the kinase two- to threefold (*West et al., 2005*; *Khan, 2005*; *Jaleel et al., 2007*). Since protein kinases are attractive pharmacological targets, this finding has raised hopes that selective LRRK2 inhibition can prevent or delay the onset of PD (*Yao et al., 2013*).

Extensive studies of LRRK2 have associated it with diverse cellular processes such as Wnt signaling, mitochondrial disease, cytoskeleton remodeling, vesicular trafficking, autophagy, and protein translation (*Taymans et al., 2015*; *Cookson, 2015*; *Schapansky et al., 2014*; *Papkovskaia et al., 2012*). Moreover, several LRRK2 substrates have been reported previously; however, evidence that they are phosphorylated by LRRK2 in a physiological context is generally lacking and proofs are confined to in vitro approaches or to cellular systems using overexpressed kinase (*Jaleel et al., 2007*; *Kumar et al., 2010*; *Ohta et al., 2011*; *Kawakami et al., 2012*; *Bailey et al., 2013*; *Martin et al., 2014*; *Qing et al., 2009*; *Chen et al., 2012*; *Gloeckner et al., 2009*; *Imai et al., 2008*; *Gillardon, 2009*; *Kanao et al., 2010*; *Matta et al., 2012*; *Xiong et al., 2012*; *Yun et al., 2013*; *Yun et al., 2015*; *Krumova et al., 2015*). Significant off-target effects for LRRK2 compounds that have been used previously further complicate interpretation of the data (*Schapansky et al., 2015*). Overall, there is little consensus on the cellular roles of LRRK2; thus, identification of definitive and verifiable physiological LRRK2 substrates is considered to be one of the greatest challenges in the field (*Schapansky et al., 2015*).

Besides mutations in LRRK2, other genetic risk variants for PD map to the Park16 locus. Among the five genes within this locus is Rab7L1 (also known as Rab29), which together with LRRK2 increases nonfamilial PD risk. Depletion of Rab7L1 recapitulates the dopaminergic neuron loss observed with LRRK2-G2019S expression and its overexpression rescues mutant LRRK2 phenotypes (*MacLeod et al., 2013*). Rab GTPases comprise ~70 family members in humans, and they are key players in all forms of intracellular vesicular trafficking events (*Stenmark, 2009*; *Rivero-Ríos et al., 2015*). Apart from Rab7L1, several other family members have been associated with PD pathogenesis. For example, mutations in Rab39b (Park21 locus) predispose to PD in humans (*Wilson et al., 2014*; *Mata et al., 2015*). Moreover, overexpression of Rab8a, Rab1, and Rab3a attenuate α-synclein-induced cytotoxicity in cellular and animal models of PD, suggesting a functional interplay

between Rab GTPases and known PD factors (*Cooper, 2006*; *Gitler et al., 2008*). Recently, another PD-connected protein kinase termed PTEN-Induce Kinase-1 (PINK1) has been reported to indirectly control the phosphorylation of a small group of Rabs including Rab8a at Ser111 (*Lai et al., 2015*). Despite these intriguing links, it is presently unclear whether LRRK2 directly or indirectly modulates Rab GTPases at the molecular level and if so, by which mechanism.

High-resolution quantitative mass spectrometry (MS) has become the method of choice for confident identification of in vitro and in vivo phosphorylation events (*Roux and Thibault, 2013*; *Lemeer and Heck, 2009*; *Olsen et al., 2006*). With current MS instrumentation, proteomics can identify tens of thousands of phosphosites (*Sharma et al., 2014*; *Mallick and Kuster, 2010*). However, challenges in the phosphoproteomic approaches are to determine functionally relevant residues from these large datasets and to establish direct kinase-substrate relationships.

As such, we complement the power of modern phosphoproteomics with parallel genetic, biochemical and pharmacological approaches to establish direct, in vivo LRRK2 substrates. Using fibroblasts derived from two different LRRK2 knock-in mouse lines we identify a subset of Rab GTPases as bona-fide LRRK2 targets. LRRK2 phosphorylates these substrates on an evolutionarily conserved residue situated in their switch II domain both in human and murine cells and in mouse brain. The phosphorylation of Rabs by LRRK2 is direct and strikingly all LRRK2 missense mutations that contribute to PD pathogenesis increase the phosphorylation of at least three Rab GTPases. Further, we establish that different PD pathogenic mutations modulate the interaction with a number of regulatory proteins including guanine dissociation inhibitors (GDI1/2). In this way, LRRK2 regulates the specific insertion of Rab GTPases into target membranes thereby altering their membrane-cytosol equilibrium.

## Results

### Identification of LRRK2 substrates in mouse embryonic fibroblasts (MEFs)

To search for bona-fide physiological LRRK2 substrates, we performed a dual-phosphoproteomic screening approach using knock-in lines harboring either hyperactive LRRK2 or a LRRK2 variant with wild-type activity but insensitive to a highly selective, newly developed LRRK2 compound. For our first screen (PS1), we generated a mouse model harboring the LRRK2-G2019S substitution that increases kinase activity two- to threefold (*Figure 1B*). We derived fibroblasts from these animals and treated them with two structurally different LRRK2 inhibitors, GSK2578215A (*Reith et al., 2012*) or HG-10-102-01 (*Choi et al., 2012*) (*Figure 1A* and *Figure 1—figure supplement 1A*). This screening modality offers three major advantages; first, increased activity of the G2019S-LRRK2 kinase amplifies the chance of finding bona-fide substrates, second, using an isogenic system excludes that measured phosphoproteome changes are due to differences in the genetic background and third, considering only the overlapping population of significantly modulated phosphopeptides of two structurally distinct inhibitors constitutes a very stringent criterion for specifically pinpointing LRRK2 substrates.

The second screen (PS2) added another layer of specificity by combining phosphoproteomics with genetics and chemical biology. For this, we used fibroblasts derived from either wt or A2016T-LRRK2 knock-in mice and treated them with the newly developed, highly potent and selective LRRK2 compound MLI-2 (*Fell et al., 2015*) The A2016T substitution does not change basal LRRK2 activity but decreases sensitivity to MLI-2 ~10-fold (*Figure 1C,D* and *Figure 1—figure supplement 1B*). At a dose of 10 nM, we observed a substantial decrease in phosphorylation of LRRK2-pS935, which is associated with LRRK2 kinase activity (*Dzamko et al., 2010*), in wt but not in A2016T cells (*Figure 1E* and *Figure 1—figure supplement 1C*). Under these conditions, the phosphoproteome of wt MEFs includes both LRRK2-specific and off-target sites, whereas A2016T (which is resistant to MLI-2) only includes off-targets. Therefore, direct quantitative comparison should reveal true LRRK2 substrates (*Figure 1C*).

Using a state-of-the-art workflow for phosphopeptide enrichment, label-free LC-MS/MS and the MaxQuant environment for stringent statistical data evaluation (*Humphrey et al., 2015*; *Cox and Mann, 2008*; *Cox et al., 2011*), we quantified over 9000 high-confidence phosphosites in each replicate in both screens (median R=0.80 and 0.89 in PS1 and PS2, respectively), (*Figure 1—figure*

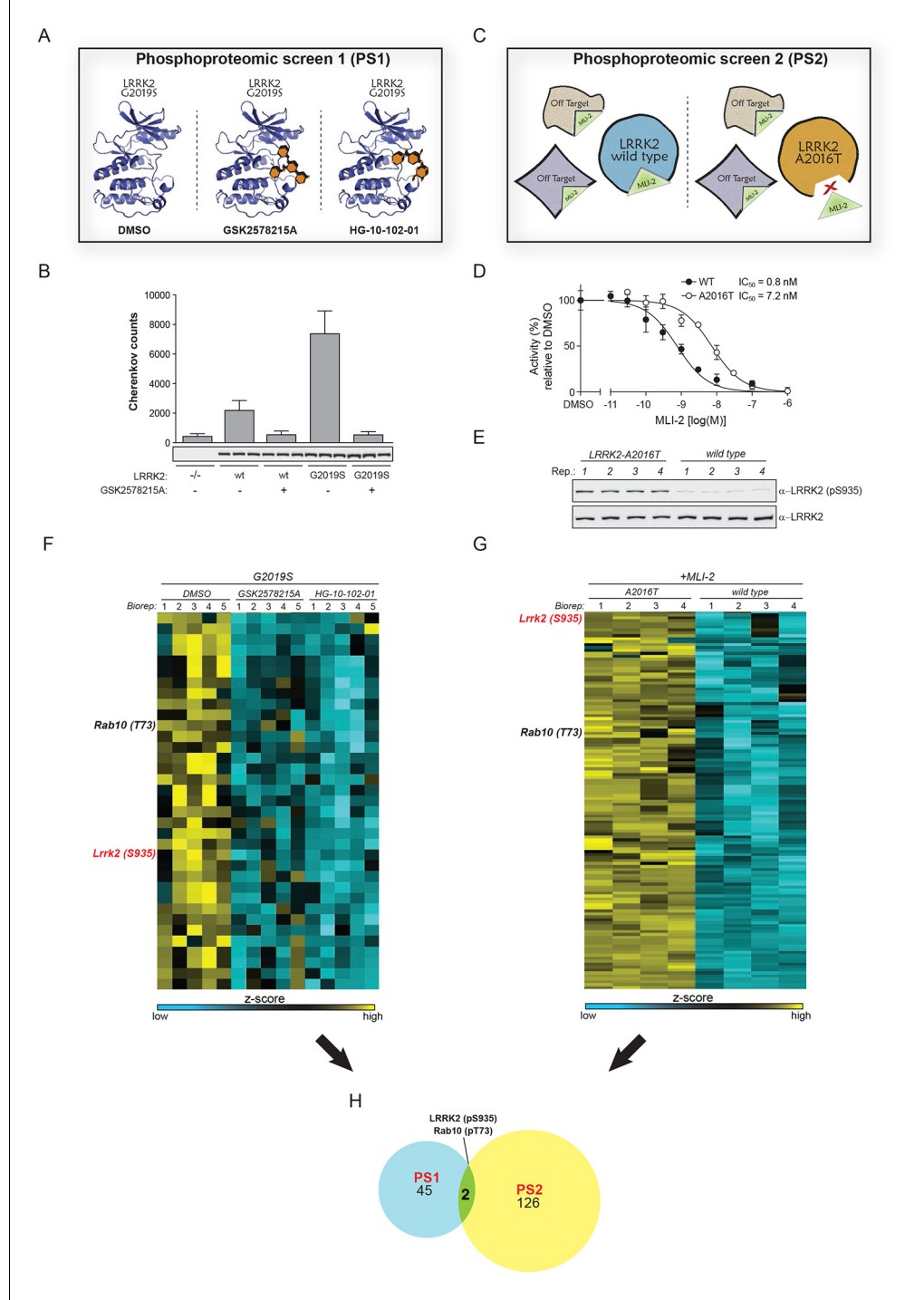

**Figure 1.** Two unbiased phosphoproteomic screens identify physiological LRRK2 targets. (**A**) Experimental setup of PS1. LRRK2-G2019S[GSK] mouse embryonic fibroblasts (MEFs, n=5) were treated with DMSO or each of two structurally distinct LRRK2 inhibitors GSK2578215A or HG-10-102-01 (1 μM for 90 min). (**B**) LRRK2 immunoprecipitated from either knockout (-/-), wild-type (wt) or LRRK2-G2019S[GSK] (G2019S) knock-in MEFs was assessed for phosphorylation of Nictide (*Nichols et al., 2009*) peptide substrate in absence or presence of GSK2578215A (2 μM). Western blot below shows that similar levels of LRRK2 were immunoprecipitated. Error bars are mean ± SD (n=3). (**C**) Scheme of PS2. The higher affinity of MLI-2 toward wt-LRRK2 allows specific pinpointing of LRRK2 substrates when comparing the phosphoproteomes of wt and A2016T MEFs. (**D**) Kinase activities of wt (closed circles) and A2016T (open circles) GST-LRRK2 [1326-2527] purified from HEK293 cells were assayed in the presence of the indicated concentration of MLI-2 (n=3). (**E**) Decreased levels of pS935-LRRK2 in wt MEFs after

*Figure 1 continued on next page*

*Figure 1 continued*

treatment with 10 nM MLI-2. (F) Heat map cluster of phosphopeptides in PS1 (p<0.005) which are downregulated after treatment with both GSK2578215A and HG-10-102-01. (G) Heat map cluster of downregulated (FDR=0.01, S0=0.2) phosphopeptides in PS2. (H) Venn diagram of overlapping downregulated phosphosites in PS1 and PS2. (Biorep= biological replicate). SD, standard deviation.

The following figure supplements are available for figure 1:

**Figure supplement 1.** Two unbiased phosphoproteomic screens identify physiological LRRK2 targets.

**Figure supplement 2.** Two unbiased phosphoproteomic screens identify physiological LRRK2 targets.

*supplement 1D–G* and *Supplementary file 1*). Independently acquired proteome measurements verified that the detected phosphorylation changes in PS2 were not due to altered protein abundances (changes as determined by label-free quantification in MaxQuant (*Cox et al., 2014*) were less than twofold [*Supplementary file 2*]).

Next, we determined how many of the identified sites were significantly and robustly modulated. As we were interested in capturing the most strongly regulated sites, we required that the fold change had to be at least as strong as pS935-LRRK2. In PS1, we thus found 234 significantly regulated sites after treatment with each of the two LRRK2 compounds GSK2578215A and HG-10-102-01 (ANOVA, p<0.005), with 78 sites regulated by both (*Figure 1—figure supplement 2A*). Hierarchical clustering divided them into several subgroups (*Figure 1—figure supplement 2B–C* and *Supplementary file 3A*). Besides revealing potential off-target sites of the two LRRK2 inhibitors, this identified a particularly interesting cluster containing 47 sites that were downregulated after treatment with both compounds (*Figure 1—figure supplement 2C*, cluster 5).

In PS2, we identified 204 significantly regulated sites (two sample t-test, FDR=0.01, S0=0.3), when comparing wild-type and inhibitor-resistant LRRK2 fibroblasts, with 128 sites specifically downregulated in the wild type, thus excluding off-target effects (*Figure 1G*, *Figure 1—figure supplement 2D* and *Supplementary file 3B*).

Finally, to stringently define LRRK2 substrates, we overlapped the results of the two orthogonal screens. Remarkably, only two phosphosites passed these stringent filtering criteria, our positive control pS935-LRRK2 and the conserved T73 residue of the small GTPase Rab10 (*Figure 1H*).

## Direct in vitro phosphorylation of Rab isoforms by LRRK2

Rab10 belongs to the Ras family of small GTPases that regulate intracellular vesicular transport, with ~70 members in human. They function as molecular switches in the tethering, docking, fusion, and motion of intracellular membranes (*Stenmark, 2009*; *Wandinger-Ness and Zerial, 2014*). The T73 residue of Rab10 is located in the switch II domain, which is characteristic of Rab GTPases (*Figure 2A*). This region changes conformation upon nucleotide binding and regulates the interaction with multiple regulatory proteins (*Pfeffer, 2005*). Sequence alignment revealed that the equivalent site to T73-Rab10 is highly conserved in more than 40 human Rab-family members, indicating strong functional relevance (*Figure 2B*). Moreover, superposing the crystal structures of multiple Rab GTPases localizes the equivalent residues to T73-Rab10 in nearly the same position (*Figure 2—figure supplement 1A*). To investigate whether the phosphorylation of Rab10 by LRRK2 is direct, we performed an in vitro kinase assay using recombinant components. Notably, we found that both wt and LRRK2-G2019S, but neither kinase inactive D1994A mutant nor small molecule-inhibited LRRK2, efficiently phosphorylated Rab10, proving a direct kinase-substrate relationship (*Figure 2C*). Furthermore, incubation of Rab10 with LRRK2 followed by tryptic digestion and MS analysis unambiguously identified T73 as the major phosphorylation site (*Figure 2—figure supplement 1B*). Given the high conservation of T73-Rab10, we investigated whether other Rab GTPases were also phosphorylated by LRRK2 in vitro. Therefore, we first measured LRRK2-mediated phosphorylation of Rab8a, Rab1a, and Rab1b, all of which contain a Thr at the site equivalent to T73-Rab10, by MS or $^{32}$P incorporation followed by Edman sequencing. Remarkably, all proteins were rapidly phosphorylated on the predicted LRRK2 phosphorylation site in the switch II domain (*Figure 2D–F* and *Figure 2—figure supplement 1C–E*). Next, we compared Rab family members containing Thr sites in

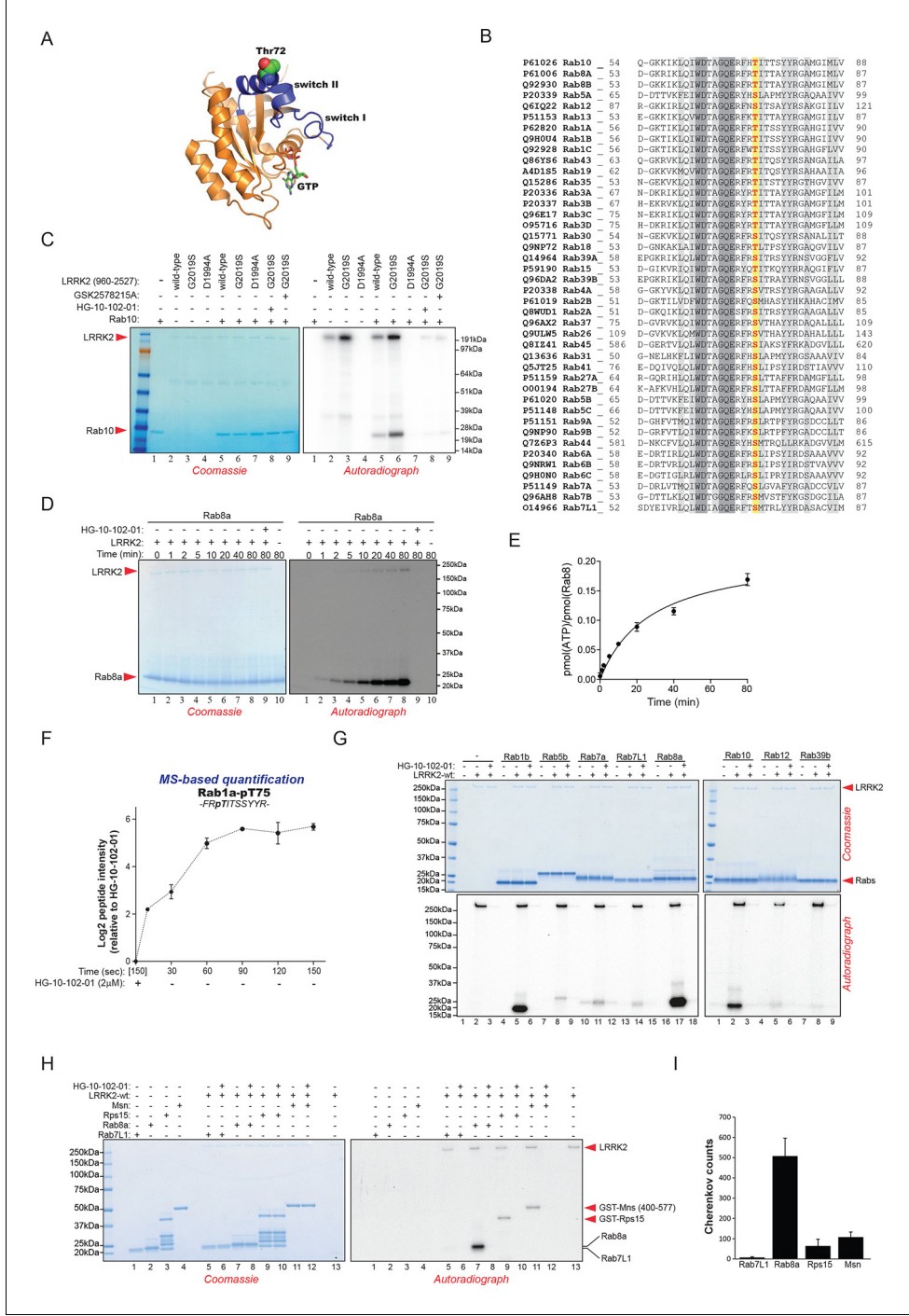

**Figure 2.** Phosphorylation of Rab GTPases by LRRK2 in vitro. (**A**) Position of threonine 72 in the switch II region of Rab8a (PDB: 4HLY). (**B**) Sequence alignment of Rab10 and other indicated Rab-family members. (**C**) Phosphorylation of Rab10 (1 µM) by wt-, G2019S- or kinase inactive LRRK2-D1994A. Inhibition of LRRK2-G2019S by GSK2578215A or HG-10-102-01 prevents phosphorylation. (**D**) Time course of LRRK2 (wt) mediated Rab8a (4 µM) phosphorylation and (**E**) quantification of phosphorylation stoichiometry (n=3). (**F**) Time course of LRRK2-wt-mediated pT75-Rab1a phosphorylation and MS-based label-free quantification (n=3). (**G**) In vitro phosphorylation of recombinant Rab proteins (4 µM) by LRRK2-wt. (**H**) Phosphorylation of recombinant Rab7L1, Rab8a, moesin, and Rps15 by LRRK2 and (**I**) quantification of the signals. For all reactions LRRK2 inhibitors= 2 µM and LRRK2= 100 ng. Error bars indicate mean ± SEM of replicates. MS, mass spectrometry; SEM, standard error of the mean; wt, wild type.

*Figure 2 continued on next page*

*Figure 2 continued*

The following figure supplement is available for figure 2:

**Figure supplement 1.** Phosphorylation of Rab GTPases by LRRK2 in vitro.

the switch II region with those containing a Ser in the equivalent position. Interestingly, while Rabs with threonines (Rab1b, Rab8a, and Rab10) were efficiently phosphorylated, those with the equivalent serine sites (Rab5b, Rab7a, Rab7L1, Rab12, and Rab39b) were phosphorylated to a drastically lower extent (*Figure 2G*). This confirms the previously reported in vitro preference for threonines by LRRK2 (*Nichols et al., 2009*). Finally, we performed a side by side comparison of the phosphorylation efficiencies of recombinant Rab8a and Rab7L1 against two previously reported substrates, moesin (Msn) and Rps15 (*Jaleel et al., 2007*; *Martin et al., 2014*). Msn is a cytoskeletal protein and a well-known in vitro LRRK2 substrate, whereas Rps15 is part of the 40S ribosomal subunit and its phosphorylation on Thr136 has been reported to regulate protein translation in *D. melanogaster* (*Martin et al., 2014*). In accordance with our previous observations (*Figure 2G*), phosphorylation levels of RAB7L1 were barely detectable, and even lower than those of Msn and Rps15. Strikingly, levels of pRab8a were about ten times higher as compared to Rps15 and Msn, two of the best in vitro LRRK2 substrates known to date, demonstrating that Rabs with Thr sites in the switch II domain are primary LRRK2 targets (*Figure 2H,I*).

## A subset of Rabs are physiological LRRK2 substrates

Because of the high conservation of T73-Rab10 (*Figure 2B*) and the ability of LRRK2 to phosphorylate multiple Rabs in vitro, we inspected our quantitative MS data further to determine whether all sequence and structurally equivalent sites are targets of LRRK2. This turned out not to be the case as pS72-Rab7a was not regulated in either of our screens. LRRK2 thus phosphorylates only a subset of Rab GTPases in mouse fibroblasts. Surprisingly, we noticed that pS105-Rab12, which is not phosphorylated by LRRK2 in vitro (*Figure 2G*), was among the significantly modulated sites in PS1 and also downregulated upon MLI-2 treatment in wt cells as compared to the inhibitor-resistant A2016T mutant in PS2 (*Figure 3A,B*). However, because of elevated intergroup variability and stringent FDR cut-offs, it was not selected in our first analysis. LRRK2 is found also in lower eukaryotes such as *C. elegans* and *D. melanogaster* (*Liu et al., 2011*) and T73-Rab10 is conserved in these organisms as well. Also, S105-Rab12 is present throughout the vertebrates (*Figure 3A,B*). We identified both pT73-Rab10 and pS105-Rab12 multiple times with high identification and phosphosite localization scores (*Supplementary file 1*) and the MS/MS fragmentation spectra of the corresponding synthetic peptides independently validated the MS results (*Figure 3—figure supplement 1A,B*). Total protein levels of Rab10 and Rab12 did not change appreciably in the A2016T knock-in model as judged by quantitative MS analysis, ruling out that that the observed phospho-level changes are due to differential protein expression (*Figure 3—figure supplement 2A*).

To extend the analysis of LRRK2-mediated phosphorylation of Rab10, we used human embryonic kidney cells harboring doxycycline-dependent gene expression of LRRK2-G2019S (HEK293-t-rex-flpIn). Expression of the kinase, treatment with either GSK2578215A or HG-10-102-01 and enrichment of Rab10 by immunoprecipitation followed by quantitative MS analysis confirmed a strong, LRRK2-dependent decrease of pT73-Rab10 peptide levels (*Figure 3—figure supplement 2B*). Polyclonal antibodies recognizing pT73-Rab10 and pS106-Rab12 (note that the equivalent site is S105 in mouse) independently verified LRRK2-dependent phosphorylation of both Rab isoforms in HEK293 cells (*Figure 3C,D*).

Next, we evaluated whether more Rab isoforms can be phosphorylated in a LRRK2-dependent manner in human cells, focusing on Rab1a, Rab3a, and Rab8a, all of which contain Thr as predicted LRRK2 phosphorylation site (*Figure 2B*). Therefore, we first ectopically expressed LRRK2 along with either Rab1a or Rab3a, in presence or absence of HG-10-102-01 and quantified pT75-Rab1a and pT86-Rab3a peptide levels by MS. Whereas T86-Rab3a is clearly a LRRK2 target, Rab1a is not, indicating that overexpression of LRRK2 is not sufficient to phosphorylate all Rabs in cells (*Figure 3—figure supplement 2C,D*). Next, we inhibited LRRK2 in HEK293-t-rex-flpIn cells expressing LRRK2-G2019S and quantified pT72-Rab8. Again, we found a strong decrease of pT72 peptide levels upon

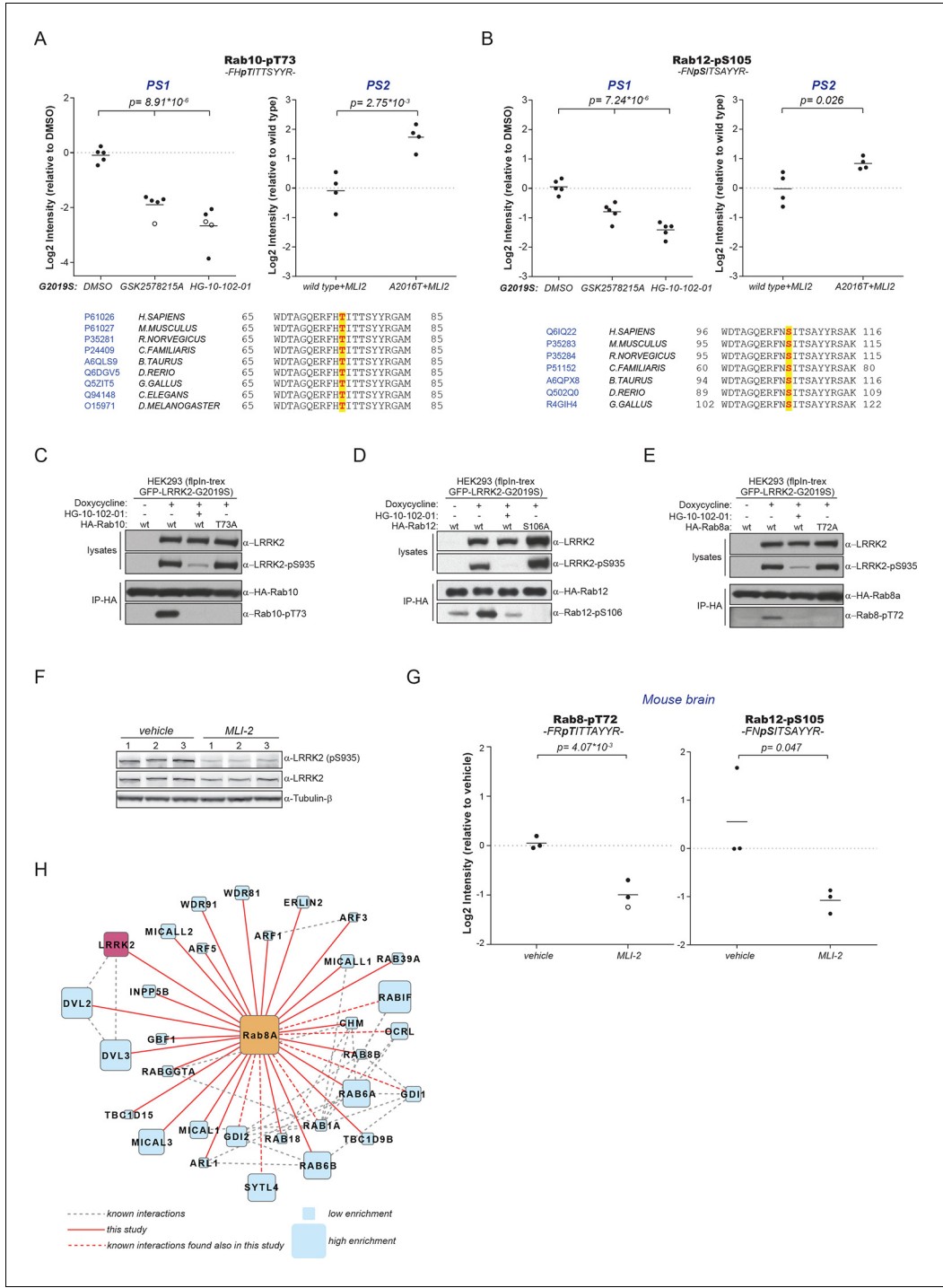

**Figure 3.** A number of Rab GTPases are physiological LRRK2 substrates. (**A**) MS-quantified pT73-Rab10 peptide intensities in PS1 and PS2. Sequence alignment of the T73-Rab10 region is shown below. (**B**) Same as (**A**) with pS106-Rab12. Western blots illustrating phosphorylation of T73-HA-Rab10 (**C**), S106-HA-Rab12 (**D**), and T72-Rab8 (**E**) after induction of LRRK2 expression by doxycycline (1 μg/ml). HG-10-102-01 (1 μM) was added prior to lysis. (**F**) Western blot of homogenized brain lysates from LRRK2-G2019S[Lilly] mice injected with vehicle (40% HPβCD) or with 3 mg/kg MLI-2 (Biorep= biological replicate) and (**G**) MS-based quantification of pT72-Rab8 and pS105-Rab12 peptides. (**H**) Cytoscape network analysis of Rab8a interacting proteins determined by affinity-purification mass spectrometry (AP-MS). LRRK2 is in purple and dashed lines in grey show experimentally determined interactions from string database (http://string-db.org/).

*Figure 3 continued on next page*

*Figure 3 continued*
The following figure supplements are available for figure 3:

**Figure supplement 1.** HCD MS/MS spectra of synthetic Rab peptides
**Figure supplement 2.** Quantification of Rab phosphorylation by mass spectrometry (MS).
**Figure supplement 3.** Several Rabs stably associate with LRRK2 in cells.

LRRK2 inhibition with both GSK2578215A and HG-10-102-01 (*Figure 3—figure supplement 2E*). An antibody raised for specific detection for pT72-Rab8 confirmed these results further (*Figure 2E*).

To analyze LRRK2-dependent phosphorylation of Rabs in an endogenous context, we quantified pT72-Rab8 and pT73-Rab10 peptide levels in MEFs derived from LRRK2 knockout, wt, or G2019S$^{GSK}$ animals. In the knock-out, the decrease was only about twofold compared to wt, implying a very low intrinsic LRRK2 activity in cells. Consistent with the two- to threefold increased in vitro activity of MEFs-extracted LRRK2-G2019S$^{GSK}$ (*Figure 1B*), our quantitative MS analysis revealed a threefold increase in both pT72-Rab8 and pT73-Rab10, which was restored to near wt levels by selective LRRK2 kinase inhibition (*Figure 3—figure supplement 2F–I*). Finally, we globally measured the brain phosphoproteome of LRRK2-G2019S$^{Lilly}$ mice injected with vehicle (40% HPβCD) or with MLI-2 (3 mg/kg). Levels of pT72-Rab8 and pS105-Rab12 were decreased more than twofold upon LRRK2 inhibition, validating our findings in the context of a LRRK2 pathogenic mouse model (*Figure 3F,G*).

Having identified Rabs as physiological substrates of LRRK2, we next asked if kinase and substrate also stably interact. Indeed, affinity-purification mass-spectrometry (AP-MS) showed that transiently expressed epitope-tagged LRRK2 and Rab8a efficiently associated with each other, demonstrating that LRRK2 is able to form stable complexes with Rab GTPases in cells (*Figure 3H* and *Figure 3—figure supplement 3A,B*). Similarly, Rab10 as well as Rab12 associate with LRRK2 when transiently overexpressed in HEK293 cells (*Figure 3—figure supplement 3C–E*).

## Parkinson's disease-associated pathogenic mutations modulate Rab GTPase phosphorylation levels

Pathogenic PD LRRK2 mutations predominantly map to the kinase and the ROC-COR (GTPase) domains and a PD risk factor coding mutation is also found in the WD-40 domain (*Martin et al., 2014*; *Farrer et al., 2007*) (*Figure 4A*). Because it is presently unclear how mutations occurring in distinct LRRK2 functional domains lead to similar disease phenotypes, we decided to investigate if different LRRK2 pathogenic mutations might impact on the phosphorylation status of Rab GTPases. For this, we expressed different disease causing LRRK2 variants along with either Rab8a or Rab10 in HEK293 cells. This revealed that besides PD-associated mutations located in the kinase domain that augment LRRK2 kinase activity, those occurring in the GTPase (ROC-COR) or the WD-40 domains also increased pT72-Rab8a and pT73-Rab10 levels in cells (*Figure 4B–E*).

To determine whether this interplay between different functional domains was direct, we next tested whether pathogenic LRRK2 mutations which lie outside the kinase domain also increase Rab phosphorylation in vitro. As expected, compared to wt, the G2019S mutation resulted in a two- to threefold increase in Rab8a phosphorylation. However, the ROC-COR domain R1441C mutation failed to do so, which is consistent with previous data suggesting that these mutations do not directly enhance LRRK2 kinase activity (*Nichols et al., 2010*), indicating that its effect on Rab8a phosphorylation levels is mediated by accessory factors in cells (*Figure 4F,G*).

## LRRK2 controls the interaction of Rabs with regulatory proteins

Rab GTPases consist of a similar core structure comprising highly conserved P-loop, switch I and switch II regions (*Pfeffer, 2005*). They cycle between the cytosol, in which they are GDP bound and inactive and specific membrane compartments, where they are activated by GDP/GTP exchange (*Hutagalung and Novick, 2011*). In the crystal structure of Rab8a (*Guo et al., 2013*), the LRRK2-mediated phosphorylation site is in the switch II region (*Figure 2A*), which regulates hydrolysis of GTP and coordinates the binding to various regulatory proteins (*Pfeffer, 2005*). We therefore tested

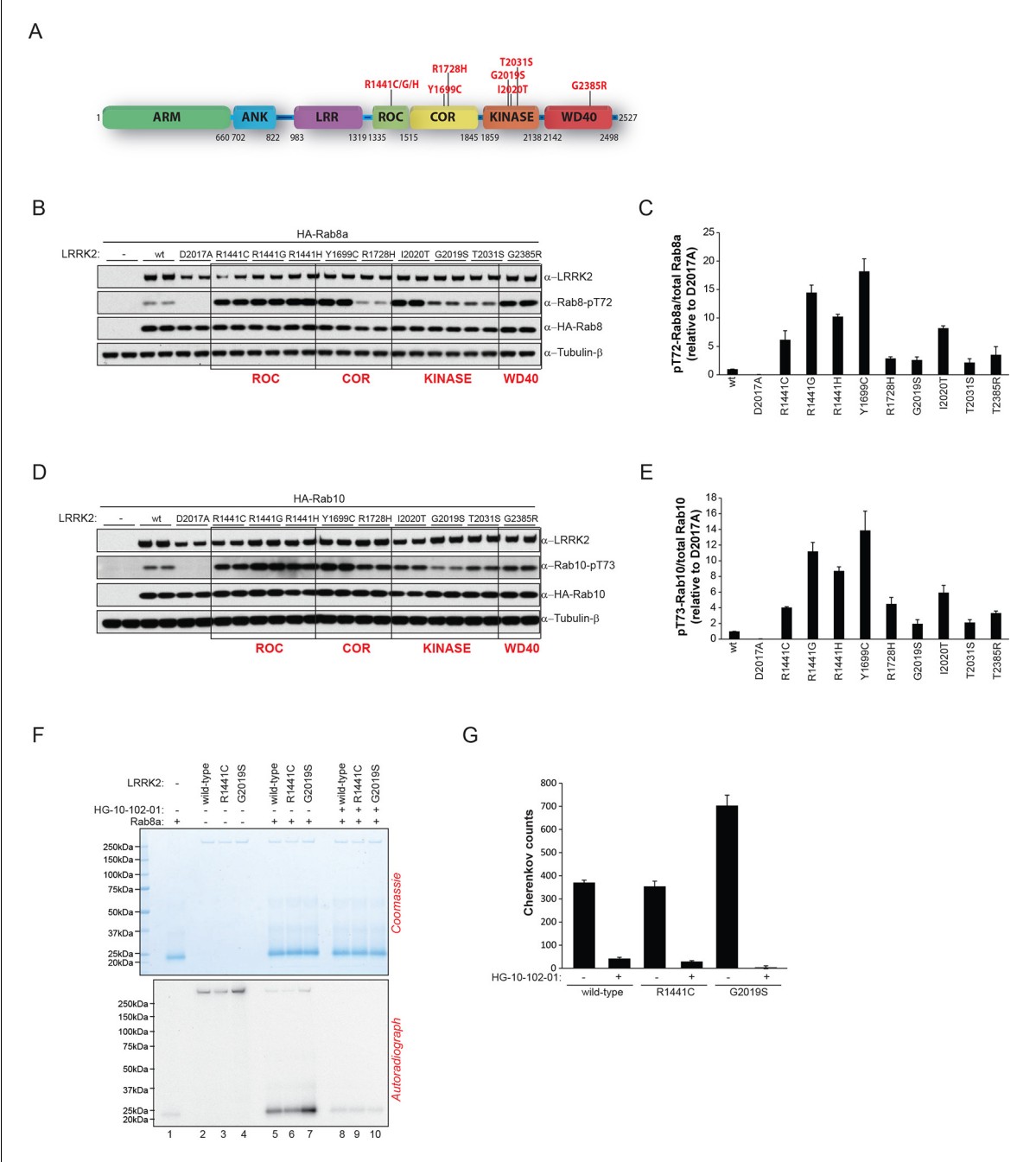

**Figure 4.** LRRK2 pathogenic variants increase phosphorylation of Rab GTPases. (**A**) Scheme of LRRK2 and common PD-associated amino acid substitutions (in red). (**B**) Different LRRK2 versions were co-expressed with Rab8a in HEK293 cells, lysates subjected to immunoblot analysis and (**C**) indicated signals quantified. (**D**) and (**E**) Same as (**B**) but HA-Rab10 was used. (**F**) In vitro phosphorylation of recombinant Rab8a (4 µM) by indicated LRRK2 variants (100 ng) and (**G**) quantification of the signals. HG-10-102-01= 2 µM. Error bars indicate mean ± SEM of replicates (n=3). PD, Parkinson's disease; SEM, standard error of the mean.

whether the phosphomimetic T72E substitution would modulate GDP/GTP binding or interfere with Rab8a protein interactions. Binding affinities of wt and the TE mutant, determined with fluorescently labeled (N-Methylanthraniloyl, mant) GDP and non-hydrolysable GTP analogue GMPPNP, did not differ (*Figure 5—figure supplement 1*). In contrast, AP-MS revealed that a number of proteins preferentially bind to non-phosphorylatable T72A-Rab8a compared to the T72E

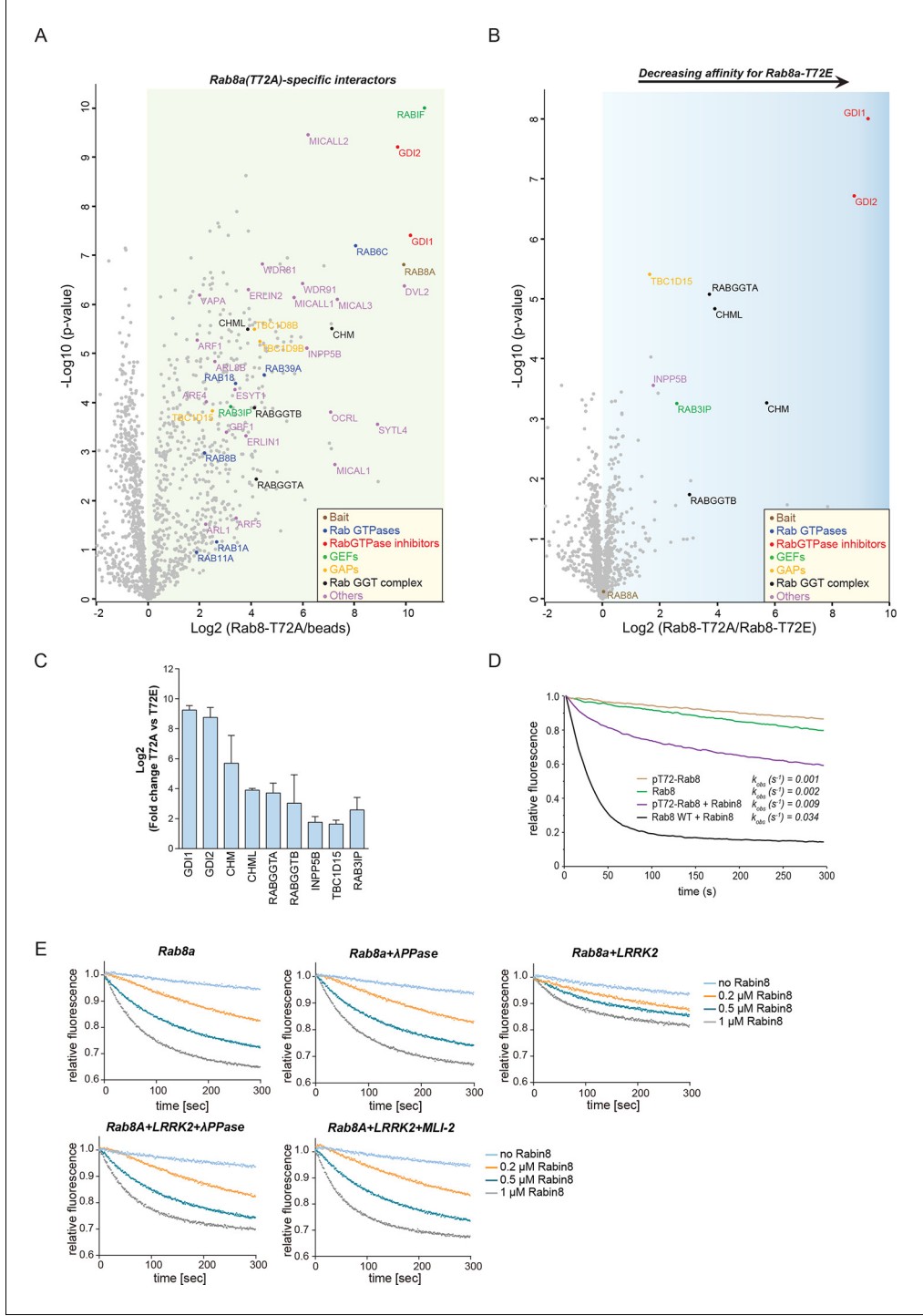

**Figure 5.** LRRK2 controls the interaction of Rabs with regulatory proteins. (**A**) Volcano plots showing interactors of GFP-Rab8a (T72A) transiently expressed in HEK293 cells and (**B**) Proteins differentially binding to T72A as compared to T72E. (**C**) Fold changes (T72A/T72E, n=4) of regulated proteins shown in (**B**). (**D**) Kinetic measurements of the dissociation of mant-GDP from non-phosphorylated and T72 phosphorylated Rab8a by Rabin8. Observed rate constants ($k_{obs}$) are indicated for each reaction and data points represent mean (n=3). (**E**) Measurements of mant-GDP dissociation from LRRK2 phosphorylated Rab8a by Rabin8 in absence or presence of λ-phosphatase (λ-PPase) or MLI-2 (1 μM). Error bars are mean ± SD of replicates.

The following figure supplements are available for figure 5:

*Figure 5 continued on next page*

*Figure 5 continued*
**Figure supplement 1.** Rab8a nucleotide binding experiments.
**Figure supplement 2.** Rab8a guanine nucleotide exchange assays.

phosphomimetic protein (*Figure 5A,B*). These were Rab GDP dissociation inhibitors α and β (GDI1 and GDI2), Rab geranyltransferase complex members (CHM, CHML, and RabGGTA/RabGGTB), the Rab8a guanine nucleotide exchange factor (GEF) Rabin8 (Rab3IP), a guanine nucleotide activating protein TBC1D15 and the inositol phosphatase INPP5B (*Figure 5C*).

Rabin8 interacts with membrane-bound Rab8a and activates it by catalyzing the exchange of GDP to GTP (*Westlake et al., 2011*). This in turn triggers retention of Rab effector proteins that mediate downstream vesicular trafficking events. Rabin8 binds to the switch II domain of Rab8a and contacts the conserved, phosphorylatable T72 residue (*Figure 5—figure supplement 2A*) (*Guo et al., 2013*). We found that compared to wt, the T72E phosphomimetic substitution decreased the level of Rabin8-catalyzed mant-GDP displacement from a Rab8-GDP complex (*Figure 5—figure supplement 2B–D*). To further substantiate this finding, we phosphorylated purified Rab8a using LRRK2, which resulted in ~60% of T72-phosphorylated protein as determined by total protein MS and LC-MS/MS after tryptic digestion (*Figure 5—figure supplement 2E,F*). Further enrichment by ion-exchange chromatography yielded a highly enriched (~100%) fraction of pT72-Rab8a (*Figure 5—figure supplement 2E*). Loading of non-phosphorylated and phosphorylated Rab8a with mant-GDP following incubation with Rabin8 revealed that LRRK2-induced phosphorylation of T72-Rab8a inhibits rates of Rabin8-catalyzed GDP exchange fourfold and decreases Rab8a-Rabin8 interaction (*Figures 5D* and *Figure 5—figure supplement 2G*). Both λ-phosphatase treatment of LRRK2-phosphorylated Rab8a and pharmacological inhibition of LRRK2 prevented the decreased GEF activity of Rabin8 toward pT72-Rab8a (*Figure 5E* and *Figure 5—figure supplement 2H*). Thus, phosphorylation of Rab8a by LRRK2 can limit its activation by Rabin8.

## PD pathogenic LRRK2 mutations interfere with Rab-GDI1/2 association

GDI1 and GDI2, along with CHM and CHML (also known as Rab escorting proteins REP1 and REP2) form the GDI superfamily and are essential regulators of the Rab cycle. GDIs extract inactive, prenylated Rabs from membranes and bind them with high affinity in the cytosol (*Pylypenko et al., 2003*). The regulatory mechanism by which Rabs are displaced from GDIs to facilitate their insertion into specific target membranes is unknown. The co-crystal structure of GDI1 with the yeast Rab homologue Ypt1 shows that GDIs closely contact the switch II region (*Rak, 2003*), which explains why phosphorylation in this domain interferes with the Rab-GDI interaction. Since GDIs are not specific to one Rab isoform (*Seabra and Wasmeier, 2004*), we reasoned that phosphorylation of the switch II domain could be a general mechanism of Rab-GDI dissociation. We therefore substituted S106-Rab12 and T73-Rab10 with non-phosphorylatable Ala or phosphomimetic Glu residues and tested their capacity to form complexes with GDIs by immunoprecipitation followed by MS or western blotting. As compared to non-phosphorylatable Rab10 and Rab12, neither S106E-Rab12 nor T73E-Rab10 was able to bind GDIs, demonstrating the functional importance of these residues (*Figure 6A,B* and *Figure 6—figure supplement 1A,B*).

To further analyze the effect of Rab phosphorylation and GDI dissociation in the context of PD, we expressed LRRK2 variants harboring various pathogenic mutations along with Rab8a in cells and assessed Rab-GDI complex formation by immunoprecipitation. Strikingly, the level of Rab8a-GDI interaction closely correlated with the degree of T72-Rab8a phosphorylation (*Figure 6C,D*). Similarly, LRRK2-mediated phosphorylation of S106-Rab12 diminished the interaction with GDIs, confirming that the effect is not specific to one Rab isoform (*Figure 6E,F*). All tested LRRK2 pathogenic mutations that affect kinase activity thus control the interaction of Rabs with GDIs. Finally, to directly test whether disruption of the Rab-GDI interaction results in an altered subcellular distribution of Rabs, we quantitatively determined T72A-Rab8 and T72E-Rab8 protein abundances in SILAC (*Ong, 2002*) labeled HEK293 cells. This revealed a twofold increase of non-phosphorylatable T72A mutant in the cytosol. Consistently, we detected a significant ($p = 2.58 \times 10^{-3}$) increase of T72E-Rab8 protein levels

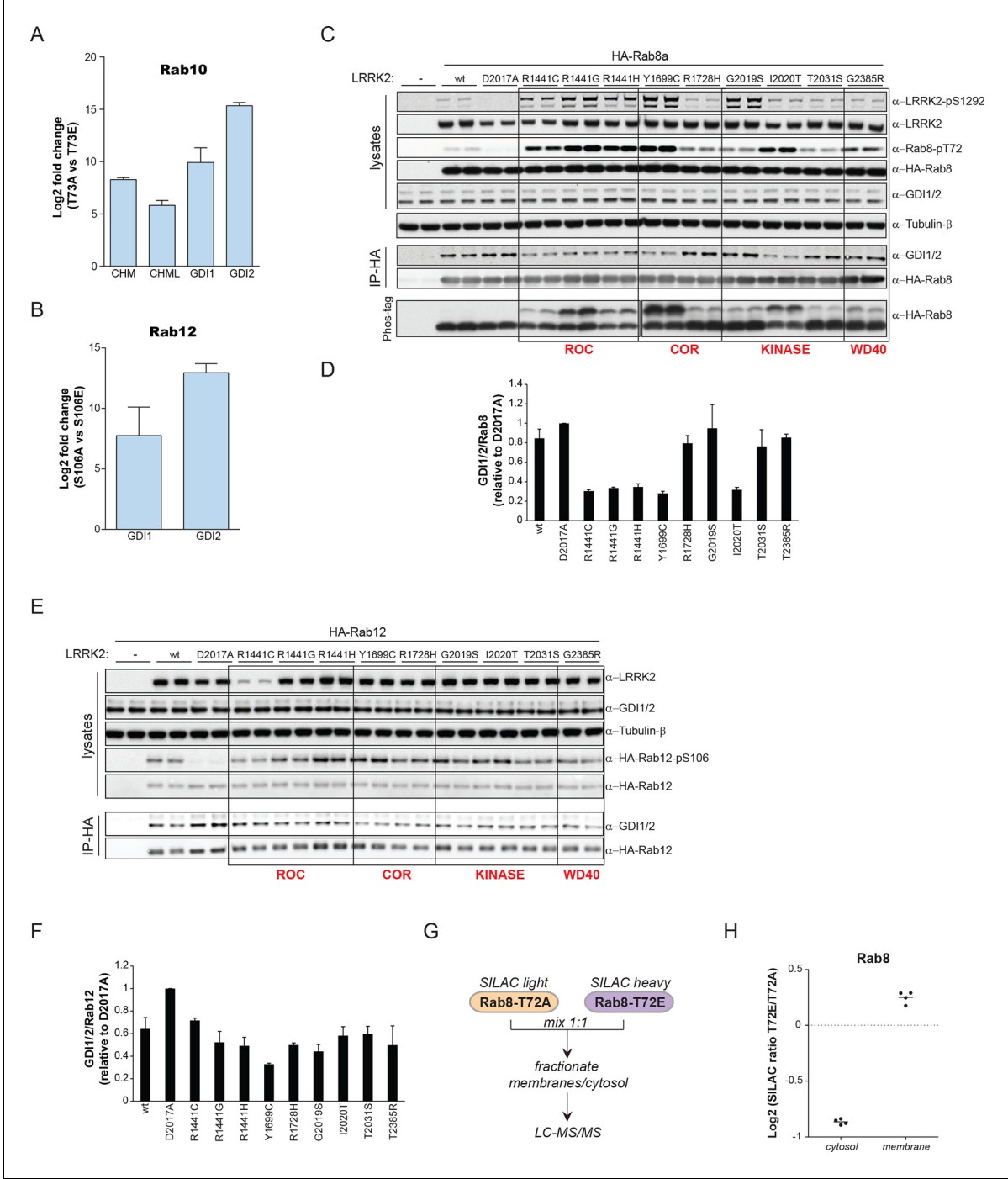

**Figure 6.** PD pathogenic LRRK2 mutations interfere with Rab-GDI1/2 association. (**A**) Fold changes (T73A/T73E, n=3) of indicated MS-quantified Rab10 interactors. (**B**) Same as (**A**) but S106A-Rab12 and S106E-Rab12 (n=4). (**C**) Different LRRK2 versions were co-expressed with Rab8a in HEK293 cells, lysates subjected to immunoblot analysis or immunoprecipitation using α-HA antibodies and indicated signals quantified (**D**). (**E**) and (**F**) Same as (**C**) with Rab12 expression. (**G**) Scheme for analyzing T72A-Rab8a and T72E-Rab8a subcellular protein distributions in a SILAC experiment. (**H**) SILAC ratios (Log2) of T72E-Rab8a/T72A-Rab8a proteins in the cytosolic and membrane fraction of HEK293 cells. PD, Parkinson's disease.

The following figure supplement is available for figure 6:

**Figure supplement 1.** Rab10/12-GDI interactions.

in the membrane fraction, demonstrating that interference with the Rab-GDI interaction results in an unbalanced membrane-cytosol distribution of Rabs (*Figure 6G,H*).

## Discussion

Here, we used a state of art MS-based phosphoproteomics workflow in combination with cells of two genetically engineered mouse models as well as a mixture of selective LRRK2 compounds to define LRRK2 targets with high stringency. Starting with almost 30,000 identified phosphosites, our screens rapidly narrowed down the candidates to a small number that were consistently and strongly regulated with all tested compounds and genetic models. Only the known phosphorylation site pS935 on LRRK2 itself and a specific residue in the Rab10 GTPase (T73) fulfilled our most specific criteria. LRRK2 kinase is conserved also in flies and worms and this is true of the T73-Rab10 substrate site as well. Further experiments with diverse model systems and techniques all verified the T73-Rab10 site as well as the equivalent sites on many but not all other Rab family members. These include the threonine sites on Rab8a and Rab3a (T72 and T86, respectively), as well as S106-Rab12. Rab7a is an important component of the endocytic pathway (*Wandinger-Ness and Zerial, 2014*) and phosphorylation on S72 has recently been shown to play a functional role in B-cell signaling (*Satpathy et al., 2015*). While our data clearly show that this site is not regulated by LRRK2 in mouse fibroblasts, its regulation by LRRK2 in B cells remains possible, given the high expression levels of LRRK2 in those cell types (*Gardet et al., 2010*). In vitro experiments proved that LRRK2 directly phosphorylates Thr but not the Ser sites in Rab isoforms, in line with its well-established in vitro preference (*Jaleel et al., 2007*; *Martin et al., 2014*; *Nichols et al., 2009*). We found that Ser sites on Rabs were hardly phosphorylated in vitro but S105-Rab12 (S106 in human) was clearly regulated in cells and brain tissue, establishing that accessory factors are required in this case. Consistent with this finding, the major characterized in vivo LRRK2 autophosphorylation site is a Ser residue (Ser1292) (*Sheng et al., 2012*). Our observation of residual Thr72-Rab8 phosphorylation in LRRK2$^{-/-}$ mice implies that one or more other kinase(s) are able to act upon this residue.

Besides defining Rab GTPases as LRRK2 targets, our screens identified a number of phosphosites as potential LRRK2 targets. However, these were validated by only one of the screens, their regulation was weaker and for many of them regulation may reflect indirect modulation by the LRRK2 kinase. This is likely to account for the difficulty in identifying substrates of this kinase. In a direct comparison of threonine Rab phosphorylation it was much stronger than the known in vitro LRRK2 targets we tested. Overall, the relatively small number of regulated sites in our screens, suggest that LRRK2 is a very specific or low activity kinase. LRRK2 is ubiquitously expressed, but highly abundant in the kidney, lungs, pancreas, and certain cell types of the immune system (*Schapansky et al., 2015*). Therefore, it is possible that different LRRK2 substrates, including Rab isoforms, are phosphorylated in a cell- or tissue-specific manner. Further phosphoproteomic research should shed more light on this open question. We conclude that the threonine sites on Rab family members identified here may not be the only functional ones in the context of LRRK2, but that they are the most prominent ones.

In searching for a functional role for LRRK2-mediated Rab phosphorylation, we noted that it maps onto the switch II region, which is known to mediate GDP/GTP exchange as well as interaction with regulatory proteins. Results from nucleotide affinity measurements make the former mechanism unlikely but AP-MS established phosphorylation-dependent binding of several proteins involved in regulating their cycling between cytosol and membrane compartments. This indicates that direct phosphorylation of Rabs on a conserved residue situated in the switch II domain regulates their movement by controlling the interaction with numerous regulatory proteins. The affinities of GDIs for Rabs are vastly decreased in a manner correlating with the phosphorylation levels induced by different LRRK2 pathogenic variants. Our data thus establish that LRRK2 is an important regulator of Rab homeostasis which is likely contributing to PD development (*Figure 7A*). Overactive LRRK2, which results in increased Rab phosphorylation, promotes dissociation from GDIs in the cytosol with concomitant membrane insertion (*Figure 7B*). In this way, the relative pool of membrane bound and cytosolic Rab is altered, disturbing intracellular trafficking. In particular, PD-associated LRRK2 mutations would shift the membrane-cytosol balance of Rabs toward the membrane compartment, thereby causing accumulation of inactive Rabs in the membranes (*Figure 7B*). The subtle increase in

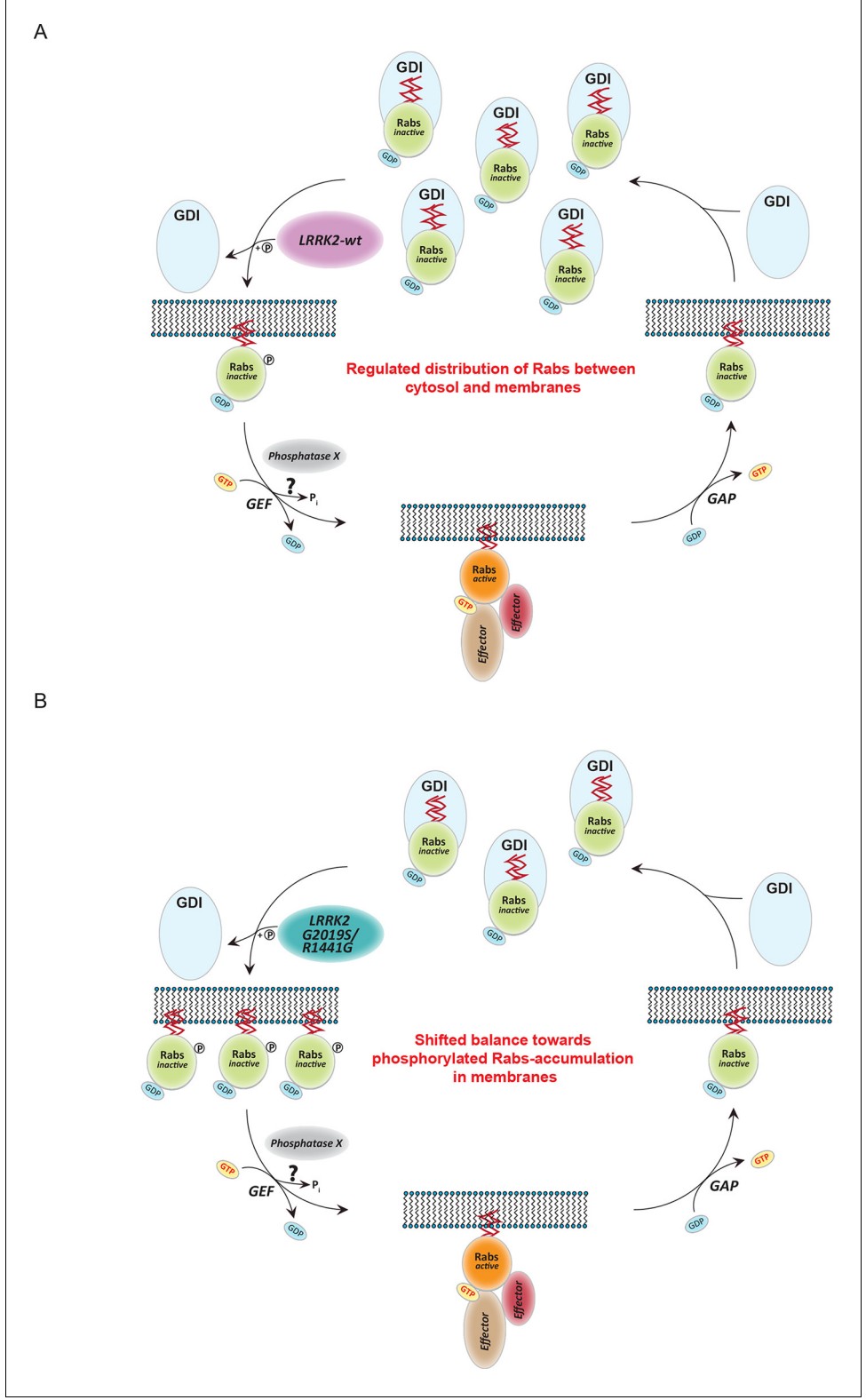

**Figure 7.** Model of Rab GTPase phosphorylation by LRRK2 and its outcome. (**A**) Rab GTPases (Rabs) cycle between an inactive (GDP-bound) and an active state (GTP-bound) between cytosol and membranes, respectively. Geranyl-geranyl-modified Rab GTPases in their GDP-bound state are tightly bound by guanine dissociation inhibitors (GDIs) in the cytosol. LRRK2 aids the insertion of Rabs in their specific target membrane. After removal of the LRRK2 phosphorylation site, guanine exchange factors (GEFs) facilitate exchange of GDP to GTP. This in

*Figure 7 continued on next page*

*Figure 7 continued*
turn allows binding to effector proteins and membrane trafficking events. Next, a Rab-specific GTPase-activating protein (GAP) assists in the hydrolysis of GTP followed by removal of the Rab GTPase from the target membrane by GDIs. (**B**) In pathogenic conditions, in which LRRK2 is hyperactive, RabGTPases have strongly diminished affinities for GDIs. As a result, the equilibrium between membrane-bound and cytosolic Rabs is disturbed, which may contribute to LRRK2 mutant carrier disease phenotypes. Model adapted and modified from (*Hutagalung and Novick, 2011*).

Rab phosphorylation in cells derived from LRRK2-G2019S knock-in mice is consistent with the long time needed for PD to manifest in humans.

Intriguingly, our results show that pathogenic LRRK2 mutations outside the kinase domain can also increase Rab phosphorylation. Although our in vitro data clearly shows that this mechanism is indirect, they will still act on the same pathway as kinase domain mutants. Therefore, the same model would also be applicable in this case.

Independent evidence that Rabs are likely to be primary LRRK2 substrates comes from LRRK2 knockout animal studies. LRRK2$^{-/-}$ mice and rats have deformed kidneys and lungs, indicative of defects in the autophagosome/lysosome pathway, which depend on properly tuned Rab activity (*Herzig et al., 2011*; *Hinkle et al., 2012*; *Baptista et al., 2013*). Moreover, it was recently reported that LRRK2 and Rab2a regulate Paneth cell function, which is compromised in Crohn's disease (*Zhang et al., 2015*). In this context, it is interesting that we found pS70-Rab2a/b to be regulated in our second screen, although it's very low abundance impedes meaningful statistical interpretation. Taken together these observations make it plausible that LRRK2 regulates Rab2a by direct phosphorylation of S70 in specialized cell types.

In conclusion, we prove that LRRK2 induces phosphorylation of Rabs and provide evidence that it deregulates cycling between cytosol and target membrane compartments. It will be interesting to investigate whether the Rab regulatory mechanism uncovered here is of key importance in vesicular trafficking in general. Discovery of a key physiological LRRK2 substrate should inform and accelerate research into PD, including monitoring the efficacy of therapeutic intervention.

## Materials and methods

### Reagents
MLI-2 (*Fell et al., 2015*) was obtained from Merck, GSK2578215A (*Reith et al., 2012*) from Tocris or GlaxoSmithKline. HG-10-102-01 was custom synthesized by Natalia Shapiro (University of Dundee) as described previously (*Choi et al., 2012*). Doxycycline, ATP, and trypsin were from Sigma. LysC was obtained from Wako. $^{32}$P-γATP was from PerkinElmer. GST-LRRK2 (residues 960-2527 wild type, G2019S, D1994A), full-length wild type flag-LRRK2 from Invitrogen and MANT-GDP (2'-(or-3')-O-(N-Methylanthraniloyl) Guanosine 5'-Diphosphate, Disodium Salt) and MANT-GMPPNP from Jena Bioscience. GFP beads for affinity purification were from Chromotek. Recombinant Rab10 and Rab1a (*Figure 3A and 3C*) were purchased from mybiosource.

### Antibodies
Anti-Rab10 and Rab8 were from Cell Signaling Technologies, anti-GFP from Invitrogen, anti-HA high affinity from Roche, anti-GDI1/2 from Sigma, and anti-pS1292-LRRK2 from Abcam. Rabbit monoclonal antibodies for total LRRK2 and pS935-LRRK2 were purified at the University of Dundee (*Dzamko et al., 2012*). Antibodies against Rab8a phospho-Thr72 (S874D), Rab10 phospho-Thr73 (S873D) and Rab12 phospho-Ser106 (S876D) were generated by injection of the KLH (keyhole limpet hemocyanin)-conjugated phospho-peptides AGQERFRpTITTAYYR (Rab8a), AGQERFHpTITTSYYR (Rab10), AGQERFNpSITSAYYR (Rab12) and IAGQERFpTSMTRLYYR (where pS/T is phospho-serine/threonine) into sheep and affinity purified using the phosphopeptides. Antibodies were used at final concentrations of 1 µg/ml in the presence of 10 µg/ml of non-phosphorylated peptide.

## Plasmids

The following constructs were used: 6His-SUMO-Rab8a wt/T72A/T72E (DU47363, DU47433, DU47436), HA-Rab8 wt/T72A/T72E (DU35414, DU47360), 6His-SUMO-Rab5b (DU26116), 6-His-SUMO-Rab7a (DU24781), 6-His-SUMO-Rab7L1 (DU50261). 6-HIS-SUMO-Rab10 (DU51062), HA-Rab10 wt/T73A/T73E (DU44250, DU51006, DU51007), 6-His-SUMO-Rab12 (DU52221), HA-Rab12 wt/S106A/S106E (DU48963, DU48966, DU48967), and 6-His-SUMO-Rab39b (DU43869). Full datasheets are available on https://mrcppureagents.dundee.ac.uk/.

## Protein purification

### Purification of Rabs and Rabin8 (*Figures 2D, 2G*, and *5E*)

Buffer A: 50 mM HEPES pH 8.0, 500 mM LiCl, 1 mM $MgCl_2$, 10 µM GDP, 2 mM beta-mercaptoethanol; Elution buffer: buffer A + 500 mM imidazole; Lysis buffer: buffer A + 1 mM PMSF and Roche Complete protease inhibitor cocktail (EDTA-free); Gel-filtration buffer: 20 mM HEPES pH 8.0, 50 mM NaCl, 1 mM $MgCl_2$, 10 µM GDP, 25 µM ATP, 2 mM DTT.

Rab isoforms and Rabin8 (153-237) were expressed as N-terminal His-Sumo fusion proteins in *E. coli* as described previously (*Bleimling et al., 2009*). The His-Sumo tag was removed using SENP1 protease (*Chaugule et al., 2011*). Transformed BL21 (DE3) harboring the GroEL/S plasmid (*Bleimling et al., 2009*) were grown at 37°C to an OD 600 of 0.8, then shifted to 19°C and protein expression induced with Isopropyl β-D-1-thiogalactopyranoside (0.5 mM) for 16 hr. Cells were pelleted at 5000 g (4°C for 20 min) and lysed by sonication (45% amplitude, 20 s pulse, 1 min pause, total of 10 min pulse) in lysis buffer. Lysates were clarified by centrifugation at 30,000 g for 20 min at 4°C followed by incubation with 1 ml of Nickel-NTA agarose/l culture for 1 hr at 4°C. The resin was equilibrated on an AKTA FPLC with buffer A, and bound proteins eluted with imidazole gradient (25 mM-500 mM). Fractions containing the protein of interest were identified by SDS-PAGE and pooled; 10 µg/ml His-SENP1 catalytic domain (residues 415–643) was used to cleave the His tag (16 hr at 4 °C). Imidazole was removed by buffer exchange gel filtration on a G25 column equilibrated in Buffer A plus 20 mM Imidazole. His-SENP1 protease was removed using Nickel-NTA agarose. Proteins were concentrated to a maximum of 10 mg/ml using a Vivaspin 10 kDa cut centricon and 0.5 ml samples were resolved on a high-resolution Superdex 200, 24 ml gel-filtration column equilibrated with gel filtration buffer and 0.2 ml fractions were collected. Peak fractions containing recombinant protein were pooled. Identity and purity of proteins were assessed by Maldi-TOF MS and SDS-PAGE.

### Purification of Rab8a and Rabin8 (*Figures 5D*)

Human Rab8a (wt, T72E, T72A, residues 1-183) and human Rabin8 constructs (144-460 and 144-245) containing tobacco etch virus (TEV) cleavable N-terminal 6-HIS tag were expressed in *E. coli* BL21 (DE3). Cells were lysed by sonication in a buffer containing 50 mM phosphate pH 7.5, 150 mM NaCl, 10% glycerol. For Rab8a proteins 5 mM $MgCl_2$ was added. Proteins were purified by Ni-NTA affinity chromatography. To remove the N-terminal HIS tag, proteins were incubated with TEV protease overnight. Further purification was done by ion-exchange chromatography (Q-Sepharose) followed by size exclusion chromatography (SEC) in a buffer of 10 mM HEPES pH 7.5, 150 mM NaCl, and 2 mM DTT using a HiLoad Superdex 75 column. For Rab8a proteins 5 mM $MgCl_2$ was added to the SEC buffer.

## Mice

Mice were maintained under specific pathogen-free conditions at the University of Dundee (UK). All animal studies were ethically reviewed and carried out in accordance with Animals (Scientific Procedures) Act 1986, the GSK Policy on the Care, Welfare and Treatment of Animals, regulations set by the University of Dundee and the U.K. Home Office. Animal studies and breeding were approved by the University of Dundee ethical committee and performed under a U.K. Home Office project license.

The LRRK2-G2019S[GSK] knock-in mouse line was generated by a targeting strategy devised to introduce the point mutation G2019S into exon 41 of the LRRK2 gene by homologous recombination in mouse embryonic stem (ES) cells. 5' and 3' homology arms (approximately 4.8 and 3.8 kb, respectively) flanking exon 41 were generated using Phusion High-Fidelity DNA Polymerase (New England BioLabs) on a C57BL/6J genomic DNA template. Similarly a 739 bp fragment carrying exon

41 lying between these two homology arms was isolated and subjected to site-directed mutagenesis with the QuickChangeII site-directed mutagenesis kit (Stratagene) to introduce the appropriate point mutation (GG to TC mutation at bps 107/8). The 5' and 3' homology arms and the mutated exon 41 fragments were subcloned into a parental targeting vector to achieve the positioning of the loxP and FRT sites and PGKneo cassette. Gene targeting was performed in de novo generated C57BL/6J-derived ES cells. The targeting construct was linearized and electroporated into ES cells according to standard methods. ES cells correctly targeted at the 3' end were identified by Southern blot analysis of *Eco*RV digested genomic DNA using a PCR-derived external probe. Correct gene targeting at the 5' end and presence of the point mutation was confirmed by sequencing of a ~6 kb PCR product. High-fidelity PCR of ES cell clone-derived genomic DNA using primers spanning the 5' homology arm generated the latter. Correctly targeted ES cell clones were injected into BALB/c blastocysts and implanted into foster mothers according to standard procedures. Male chimaeras resulting from the G2019S targeted ES cells were bred with C57BL/6J female mice, and germline transmission of the targeted allele was confirmed by PCR. The PGKneo cassette was subsequently removed by breeding germline mice to FLPeR (*Farley et al., 2000*) mice expressing Flp recombinase from the Rosa26 locus (C57BL/6J genetic background). Absence of the PGKneo cassette in offspring was confirmed by PCR and subsequent breeding to C57BL/6J mice removed the Flper locus (confirmed by PCR). The line was maintained by breeding with C57BL/6J, and crossing mice heterozygous for the point mutation generated homozygous mice. Standard genotyping which distinguishes wild type from point mutation knock-in alleles was used throughout. Requests for LRRK2 G2019SGSK mice should be directed to: alastair.d.reith@gsk.com.

The Michael J. Fox Foundation for Parkinson's Research generated the A2016T knock-in mice. A targeting vector was designed to introduce an alanine to threonine (A2016T) substitution at codon 2016 in exon 41 of the endogenous locus. In addition, an FRT-flanked neomycin resistance (neo) cassette was introduced 400 bp downstream of exon 41. The construct was electroporated into C57BL/6N-derived JM8 ES cells. Correctly targeted ES cells were injected into blastocysts and chimeric mice were bred to B6.Cg-Tg (ACTFLPe)9205Dym/J (JAX stock No. 005703) to remove the neo cassette and leave a silent FRT site. The resulting animals were crossed to C57BL/6NJ inbred mice (JAX stock No. 005304) for one generation. These mice are available from The Jackson Laboratory and for further information see http://jaxmice.jax.org/strain/021828.html. The LRRK2-G2019S[Lilly] were generated by Ely Lilly and maintained on a C57BL/6J background.

Genotyping of mice was performed by PCR using genomic DNA isolated from ear biopsies. For LRRK2-G2019S[GSK] knock-in mice, Primer 1 (5'-CCGAGCCAAAAACTAAGCTC -3') and Primer 2 (5'-CCATCTTGGGTACTTGACC-3') were used to detect the wild-type and knock-in alleles. For LRRK2-G2019S[Lilly] knock-in mice Primer 1 (5'-CATTGCGAAGATTGCGGACTACTCAATT-3') and Primer 2 (5'-AAACAGTAACTATTTCCGTCGTGATCCG-3') were used to detect the wild-type and knock-in alleles. For LRRK2-A2016T Primer 1 (5'-TTGCCTGTGAGTGTCTCTGG-3') and Primer 2 (5'-AGGAAATGTGGTTCCGACAC-3') were used to detect the wild-type and knock-in alleles. The PCR program consisted of 5 min at 95°C, then 35 cycles of 30 s at 95°C, 30 s at 60°C and 30 s at 72°C, and 5 min at 72°C. DNA sequencing was used to confirm the knock-in mutation and performed by DNA Sequencing & Services (MRC–PPU; http://www.dnaseq.co.uk) using Applied Biosystems Big-Dye version 3.1 chemistry on an Applied Biosystems model 3730 automated capillary DNA sequencer.

For experiments shown in *Figure 3F–G*, homozygous LRRK2-G2019S[Lilly] mice (3 months of age) were injected subcutaneously with vehicle (40% Hydroxypropyl-β-Cyclodextran) or MLI-2 (3 mg/kg of body mass dissolved in 40% Hydroxypropyl-β-Cyclodextran) and euthanized by cervical dislocation 1 hr after treatment. Brains were rapidly isolated and snap frozen in liquid nitrogen. No specific randomization method or blinding was applied to experiments.

## Generation of MEFs

Littermate matched wild type and homozygous LRRK2-A2016T mouse embryonic fibroblasts (MEFs) were isolated from mouse embryos at day E12.5 resulting from crosses between heterozygous LRRK2-A2016T/WT mice using a previously described protocol (*Wiggin et al., 2002*). Cells were genotyped as described above for mice and wild type and homozygous A2016T knock-in cells generated from the same littermate selected for subsequent experiments. Cells cultured in parallel at passage 4 were used for MS and immunoblotting experiments.

Littermate matched wild type and homozygous LRRK2-G2019S[GSK] MEFs were isolated from mouse embryos at day E12.5 resulting from crosses between heterozygous LRRK2-G2019S[GSK]/WT mice as described previously (*Wiggin et al., 2002*). Wild-type and homozygous LRRK2-G2019S[GSK]/ G2019S[GSK] MEFs were continuously passaged in parallel for at least 15 passages before being used for MS and immunoblotting experiments. All cells were cultured in DMEM containing 10% FBS, 2 mM L-glutamine, 50 units/ml penicillin, 50 µg/ml streptomycin, and non-essential amino acids (Life Technologies). Littermate matched wild type and homozygous knock-out MEFs were isolated from LRRK2 knock-out mice (*Dzamko et al., 2012*) as described previously (*Davies et al., 2013*). All knock-in and knock-out cell lines were verified by allelic sequencing.

## Culture and transfection of cells

HEK293 were purchased from ATCC and cultured in Dulbecco's modified Eagle medium (Glutamax, Gibco) supplemented with 10% fetal calf serum, 100 U/ml penicillin, and 100 µg/ml streptomycin. The HEK293-t-rex-flpIn stable cell lines with doxycycline-inducible wild type and mutant forms of LRRK2 have been described previously (*Nichols et al., 2010*). Transient transfections were performed 36-48 hr prior to cell lysis using Lipofectamine 2000 (Life Technologies) or FuGene HD (Promega). LRRK2 expression in HEK293-t-rex-flpIn was induced by doxycycline (1 µg/ml, 24 hr). All cells were tested for mycoplasma contamination and overexpressing lines were verified by Western blot analysis.

## Immunoprecipitations, pull-downs, and subcellular fractionation

For HA-Rab immunoprecipitations, HA-agarose (Sigma) was washed 3 times with PBS and incubated with lysates at a concentration of 25 µl of resin/mg lysate for 1 hr. Beads were then washed twice with 1 ml PBS and samples eluted in 2 x LDS (50 µl per 25 µl of resin) and centrifuged through a 0.22 µm Spinex filter, 2-mercpatoethanol added to 2% (v/v) and heated to 70°C for 5 min prior to SDS-PAGE. For GFP pulldowns and immunoprecipitations, cells were lysed in ice-cold NP-40 extraction buffer (50 mM Tris-HCl, pH 7.5, 120 mM NaCl, 1 mM EDTA, 6 mM EGTA, 15 mM sodium pyrophosphate and 1% NP-40 supplemented with protease and phosphatase inhibitors (Roche) and clarified by centrifugation at 14,000 rpm. Supernatants were incubated over night with Rab8 or Rab10 antibodies and bound complexes recovered using agarose protein A/G beads (Pierce). For GFP pull-downs, lysates were incubated with GFP beads for 2 hr (Chromotek). On bead digestion of protein complexes used for MS analysis was performed as described previously (*Hubner et al., 2010*).

For subcellular fractionation, SILAC (*Ong, 2002*) labeled HEK293 cells were counted and mixed in a 1:1 ratio after harvesting in PBS, spun at 1000 rpm at 4°C for 5 min and then resuspended in sub-cellular fractionation buffer (250 mM sucrose, 20 mM HEPES pH 7.4, 10 mM KCl, 1.5 mM $MgCl_2$, 1 mM EDTA, 1 mM EGTA and protease/phosphatase inhibitor cocktail [Roche]). Cells were then Dounce homogenized, left on ice for 20 min and spun at 750 g for 5 min. The supernatant spun in an ultracentrifuge (100,000 *g*) for 45 min to obtain cytosolic (supernatant) and membrane (pellet) fractions.

## Phos-tag SDS-PAGE

Phos-tag acrylamide and $MnCl_2$ were added to a standard gel solution at a final concentration of 50 µM and 100 µM, respectively. After degassing for 10 min, gels were polymerized by ammonium persulfate and TEMED. Cell lysates used for Phos-tag SDS-PAGE were supplemented with $MnCl_2$ at 10 mM to mask the effect of EDTA in the lysates. After SDS-PAGE, gels were washed 3 times with transfer buffer containing 10 mM EDTA followed by a wash with transfer buffer (10 min each). Blotting to nitrocellulose membranes was carried out according to a standard protocol.

## Total proteome and phosphoproteome sample preparation and MS analyses

All samples were lysed in SDS lysis buffer (4% SDS, 10 mM DTT, 10 mM Tris pH 7.5), boiled and sonicated, and precipitated overnight using ice-cold acetone (v/v= 80%). After centrifugation (4000 g), the pellet was washed at least twice with 80% ice-cold acetone before air drying and resuspension (sonication) in either urea (6 M urea, 2 M thiourea, 50 mM Tris pH 8) or TFE buffer (10% 2-2-2-

trifluorethanol, 100 mM ammonium bicarbonate [ABC]). Proteins were digested using LysC and trypsin (1:100), overnight at 37°C. Peptides for total proteome measurements were desalted on C18 StageTips and phosphopeptides were enriched as described previously (*Humphrey et al., 2015*).

## LC-MS/MS measurements

Peptides were loaded on a 50 cm reversed phase column (75 μm inner diameter, packed in-house with ReproSil-Pur C18-AQ 1.9 μm resin [Dr. Maisch GmbH]). Column temperature was maintained at 50°C using a homemade column oven. An EASY-nLC 1000 system (Thermo Fisher Scientific) was directly coupled online with a mass spectrometer (Q Exactive, Q Exactive Plus, Q Exactive HF, LTQ Orbitrap, Thermo Fisher Scientific) via a nano-electrospray source, and peptides were separated with a binary buffer system of buffer A (0.1% formic acid [FA]) and buffer B (80% acetonitrile plus 0.1% FA), at a flow rate of 250nl/min. Peptides were eluted with a gradient of 5-30% buffer B over 30, 95, 155, or 240 min followed by 30-95% buffer B over 10 min, resulting in approximately 1, 2, 3, or 4 hr gradients, respectively. The mass spectrometer was programmed to acquire in a data-dependent mode (Top5–Top15) using a fixed ion injection time strategy. Full scans were acquired in the Orbitrap mass analyzer with resolution 60,000 at 200 m/z (3E6 ions were accumulated with a maximum injection time of 25 ms). The top intense ions (N for TopN) with charge states $\geq$2 were sequentially isolated to a target value of 1E5 (maximum injection time of 120 ms, 20% underfill), fragmented by HCD (NCE 25%, Q Exactive) or CID (NCE 35%, LTQ Orbitrap) and detected in the Orbitrap (Q Exactive, R= 15,000 at m/z 200) or the Ion trap detector (LTQ Orbitrap).

## Data processing and analysis

Raw MS data were processed using MaxQuant version 1.5.1.6 or 1.5.3.15 (*Cox and Mann, 2008*; *Cox et al., 2011*) with an FDR < 0.01 at the level of proteins, peptides and modifications. Searches were performed against the Mouse or Human UniProt FASTA database (September 2014). Enzyme specificity was set to trypsin, and the search included cysteine carbamidomethylation as a fixed modification and N-acetylation of protein, oxidation of methionine, and/or phosphorylation of Ser, Thr, Tyr residue (PhosphoSTY) as variable modifications. Up to two missed cleavages were allowed for protease digestion, and peptides had to be fully tryptic. Quantification was performed by MaxQuant, 'match between runs' was enabled, with a matching time window of 0.5-0.7 min. Bioinformatic analyses were performed with Perseus (www.perseus-framework.org) and Microsoft Excel and data visualized using Graph Prism (GraphPad Software) or R studio (https://www.rstudio.com/). Hierarchical clustering of phosphosites was performed on logarithmized (Log2) intensities. Significance was assessed using one sample t-test, two-sample student's t-test, and ANOVA analysis, for which replicates were grouped, and statistical tests performed with permutation-based FDR correction for multiple hypothesis testing. Missing data points were replaced by data imputation after filtering for valid values (all valid values in at least one experimental group). Error bars are mean ± SEM or mean ± SD. Proteomics raw data have been deposited to the ProteomeXchange Consortium (http://proteomecentral.proteomexchange.org) via the PRIDE partner repository with the data set identifier PXD003071.

## LRRK2 inhibitor IC50 LRRK2 kinase assay

LRRK2 kinase activity was assessed in an in vitro kinase reaction performed as described previously (*Nichols et al., 2009*). For IC$_{50}$ determination of LRRK2 inhibitor, peptide kinase assays were set up in a total volume of 30 μl with recombinant wild type GST-LRRK2-(1326-2527) or mutant GST-LRRK2 [A2016T]-(1326-2527) (6 nM) in 50 mM Tris-HCl pH 7.5, 0.1 mM EGTA, 10 mM MgCl$_2$, 0.1 mM [γ-32P]ATP (~300-600 cpm/pmol), and 20 μM Nictide LRRK2 substrate peptide substrate, in the presence of indicated concentration of MLI-2. After incubation for 20 min at 30°C, reactions were terminated by applying 25 μl of the reaction mixture onto P81 phosphocellulose papers and immersion in 50 mM phosphoric acid. After extensive washing, reaction products were quantified by Cerenkov counting. IC50 values were calculated using non-linear regression analysis using GraphPad Prism (GraphPad Software). The IC$_{50}$s for GSK2578215A (*Reith et al., 2012*) against wild type GST-LRRK2-(1326-2527) or mutant GST-LRRK2[A2016T]-(1326-2527) are 10.9 nM and 81.1 nM, respectively, and the IC50s for HG-10-102-01 (*Choi et al., 2012*) vs LRRK2 WT and A2016T are 20.3 nM and 153.7 nM, respectively. The IC$_{50}$ of MLI-2 against wild type GST-LRRK2-(1326-2527) or mutant GST-LRRK2

[A2016T]-(1326-2527) are 0.8 nM and 7.2 nM (see Extended Data *Figure 1A*). As MLI-2 displayed greater potency as well as a higher degree of resistance between wild type and A2016T mutation (9-fold compared to 7.4-fold for GSK2578215A and 7.6-fold for HG-10-102-01), we used MLI-2 for MS studies employing LRRK2[A2016T] knock-in MEFs.

## In vitro kinase assays

### Phosphorylation of Rab isoforms (*Figures 2D, 2G, 2H* and *4F*)

LRRK2 kinase assays were performed using purified recombinant GST-tagged LRRK2 (960-2527, wt, D1994A, G2019S, Invitrogen) incubated with recombinant Rab isoform (1 µM). Proteins were incubated in kinase assay buffer (20 mM Tris pH 7.5, 1 mM EGTA, 5 mM β-glycerophosphate, 5 mM NaF, 10 mM MgCl$_2$, 2 mM DTT, 10 µM ATP, and 0.5 µCi of γ-32P-ATP) at a combined volume of 30 µL. The reaction mixture was incubated at 30°C for 30 min, or as indicated. Reactions were quenched by the addition of SDS-sample loading buffer, heated to 70°C for 10 min and then separated on SDS-PAGE. Following electrophoresis, gels were fixed (50% (v/v) methanol, 10% (v/v) acetic acid), stained in Coomassie brilliant blue, dried and exposed to a phospho-imaging screen for assessing radioactive $^{32}$P incorporation. For MS analysis, 100 µM of ATP was used and the reaction was stopped by addition of 2 M urea buffer (2 M urea in 50 mM Tris pH 7.5) containing LRRK2 inhibitor HG-10-102-01 (2 µM).

### Phosphorylation of Rab isoforms (*Figures 2C and 2F*)

Assays were set up in a total volume of 25 µl with recombinant full-length Flag-LRRK2 (100 ng, Invitrogen) in 50 mM Tris-HCl pH 7.5, 0.1 mM EGTA, 10 mM MgCl$_2$, 0.1 mM [γ-$^{32}$P]ATP (~300-600 cpm/pmol), with recombinant Rab isoform (1.5 µg) in the presence or absence of the LRRK2 inhibitor HG-10-102-01 (2 µM). After incubation for 30 min at 30°C, reactions were stopped by the addition of Laemmli sample buffer and reaction products resolved on SDS-PAGE. The incorporation of phosphate into protein substrates was determined by autoradiography. For phosphorylation of Rab8a or Nictide (*Figure 1B*) LRRK2 was immunoprecipitated from MEFs and kinase activity was assessed in an in vitro kinase reaction as described previously (*Davies et al., 2013*).

## Phosphosite identification by MS and Edman sequencing (*Figure 2—figure supplement 1*)

Purified Rab1b and Rab8a (5 µg) were phosphorylated using recombinant full-length wild type Flag-LRRK2 (0.2 µg; Invitrogen) in a buffer containing 50 mM Tris-HCl pH 7.5, 0.1 mM EGTA, 10 mM MgCl2, 0.1 mM [γ-32P]ATP (~3000 Ci/pmol) for 1 hr at 30°C. The reactions were stopped by the addition of SDS sample buffer, and reaction products were resolved by electrophoresis on SDS-PAGE gels that were stained with Coomassie blue. The band corresponding to Rab1b/Rab8a was excised and digested overnight with trypsin at 30°C, and the peptides were separated on a reverse-phase HPLC Vydac C18 column (Separations Group) equilibrated in 0.1% (v/v) trifluoroacetic acid, and the column developed with a linear acetonitrile gradient at a flow rate of 0.2 ml/min. Fractions (0.1 ml each) were collected and analyzed for $^{32}$P radioactivity by Cerenkov counting. Phosphopeptides were analyzed by liquid chromatography (LC)-MS/MS using a Thermo U3000 RSLC nano liquid chromatography system (Thermo Fisher Scientific) coupled to a Thermo LTQ-Orbitrap Velos mass spectrometer (Thermo Fisher Scientific). Data files were searched using Mascot (www.matrixscience.com) run on an in-house system against a database containing the appropriate Rab sequences, with a 10 ppm mass accuracy for precursor ions, a 0.6 Da tolerance for fragment ions, and allowing for Phospho (ST), Phospho (Y), Oxidation (M), and Dioxidation (M) as variable modifications. Individual MS/MS spectra were inspected using Xcalibur 2.2 (Thermo Fisher Scientific), and Proteome Discoverer with phosphoRS 3.1 (Thermo Fisher Scientific) was used to assist with phosphosite assignment. The site of phosphorylation of 32P-labeled peptides was determined by solid-phase Edman degradation on a Shimadzu PPSQ33A Sequencer of the peptide coupled to Sequelon-AA membrane (Applied Biosystems) as described previously (*Campbell and Morrice, 2002*).

## Phosphorylation of Rab8a (*Figure 5D*)

Rab8a was phosphorylated using LRRK2-G2019S (see 'in vitro kinase assays' section). Non-phosphorylated and phosphorylated Rab8a proteins were separated using ion-exchange chromatography

(Mono S 4.6/100; GE Healthcare) with a linear salt gradient from buffer A (20 mM Tris/HCl pH 7.5, 50 mM NaCl, 10% (v/v) glycerol) to buffer B (as buffer A, but with 1000 mM NaCl). The successful enrichment of phosphorylated Rab8a was confirmed by ESI-TOF MS.

## Rab8a nucleotide binding experiments

Rab8a (1-183, wt and T72E) were subjected to HPLC revealing that the purified proteins were (>90%) in the nucleotide-free form. To determine affinities for G-nucleotides, fluorescence measurements were carried out at 20°C in a buffer containing 50 mM Tris pH 7.5, 100 mM NaCl, and 5 mM $MgCl_2$. Spectra were measured with a PerkinElmer LS50B fluorescence spectrophotometer; 1 μM of methylanthraniloyl (mant) labeled GMPPNP and GDP was incubated with increasing concentrations of wild type and T72E Rab8a in 60 μl volumes. Fluorescence of mant-nucleotides was excited at 355 nm and emission spectra monitored from 400 to 500 nm, with emission maxima detected at 448 nm. Intrinsic protein fluorescence and mant-nucleotide background fluorescence was subtracted from the curves. Data collection was performed with the program FL WinLab (PerkinElmer), while further analysis, curve fitting and dissociation constant ($K_d$) determination was done using GraphPad Prism (GraphPad Software).

## Guanine exchange factor (GEF) assays

*Figure 5E*: Purified Rab8a (100 μg) was phosphorylated using LRRK2 G2019S (1.5 μg) in a buffer containing 50 mM Tris-HCl pH 7.5, 0.1 mM EGTA, 10 mM $MgCl_2$, 2 mM DTT, 1 mM ATP (18 hr, room temperature [RT]) in a Dispo-Biodialyzer MWCO 1 kDa (Sigma-Aldrich) and incubated in 2 l of the same Buffer to allow for ADP exchange. The buffer was subsequently exchanged to a GDP dissociation assay buffer containing 20 mM HEPES-NaOH pH 7.5, 50 mM NaCl, 2 mM DTT, 1 mM $MnCl_2$, 0.01% (w/v) Brij-35 using Zeba Spin desalting columns (Invitrogen). Phosphorylated Rab8a (50 μg) was treated with lambda phosphatase (5 μg) for 30 min at 30°C where indicated. To load mant-GDP, Rab8a was incubated with 40 μM mant-GDP in the presence of 5 mM EDTA at 30°C for 30 min. After adding $MgCl_2$ at 10 mM, in order to remove unbound mant-GDP, the buffer was exchanged to a buffer containing 10 mM HEPES-NaOH pH 7.5, 50 mM NaCl, 5 mM DTT, 1 mM $MgCl_2$ using Zeba Spin desalting columns. GDP dissociation reactions were set up in a total volume of 50 μl with 1 μM Rab8a:mant-GDP in 20 mM HEPES-NaOH pH 7.5, 50 mM NaCl, 2 mM DTT, 1 mM $MgCl_2$, and 0.1 mM GDP, and the reaction was started by adding the indicated concentration of Rabin8 (residues 153-237) (*Guo et al., 2013*). Kinetic measurement of the mant fluorescence was carried out in a black half-area 96-well plate with PHERAStar FS (BMG Labtech) at RT using a set of filters (excitation: 350 nm, emission: 460 nm). The observed rate constant ($k_{obs}$) and the catalytic efficiency ($k_{cat}/K_m$) were calculated as described previously (*Delprato et al., 2004*). Phosphorylation stoichiometry (63%) was calculated by digestion of the protein with trypsin and analyzing the fragments by Orbitrap MS.

*Figure 5D*: Phosphorylated Rab8a was obtained as described in section 'Phosphorylation of Rab8a'. GEF assays were performed as described previously (*Eberth and Ahmadian, 2009*). Loading of purified nucleotide-free (both phosphorylated and non-phosphorylated) Rab8a (1-183) with 2'(3')-O-(N-methylanthraniloyl)-GDP (mantGDP) was achieved by incubation with an 1.5 molar excess of mantGDP for 2 hr at RT. Unbound mantGDP was removed using a size-exclusion chromatography column. (Micro Bio-Spin column, Bio-RAD). The nucleotide exchange reactions were set up in a total volume of 50 μl in a quartz-glass cuvette (Hellma Analytics) with 0.5 μM mantGDP-bound Rab8a (non-phosphorylated or phosphorylated) using a GEF buffer containing 30 mM Tris pH 7.5, 5 mM $MgCl_2$, 3 mM DTT and 10 mM $KH_2PO_4$, pH 7.4. Purified Rabin8 (144-245, GEF domain) was subjected to size exclusion chromatography prior to GEF activity assay to ensure no loss of GEF activity due to storage. Rabin8 was added to a final concentration of 2 μM and incubated for 30 min at 20°C. The reactions were initiated by addition of GTP (1 mM $c_f$). The dissociation of mant-GDP from Rab8a was monitored every 2 s for a total of 300 s at 20°C using a fluorescence spectrometer (PerkinElmer, 366 nm excitation and 450 nm emission). The observed rate constants ($k_{obs}$) were calculated by fitting the data into a one-phase exponential decay equation without constraints using nonlinear regression in GraphPad Prism (GraphPad Software Inc).

## Ni²⁺-NTA Rabin8 pull-down

$Ni^{2+}$-NTA beads were pre-equilibrated with buffer containing PBS pH 7.4, 30 mM imidazole and 5 mM $MgCl_2$. Purified HIS-tagged Rabin8 (residues 144-460) and untagged Rab8a (1-183) WT or quantitatively phosphorylated pT72 were mixed at equal molar ratios. Individual proteins and a mixture of the proteins were incubated with $Ni^{2+}$-NTA beads for 1.5 hr at 4°C. Beads were washed 3 times with PBS, bound proteins eluted with 500 mM imidazole followed by SDS–PAGE and western blot analysis.

## Acknowledgements

This work was supported by the Max-Planck Society for the Advancement of Science, The Michael J. Fox Foundation for Parkinson's Research (grant ID 6986), the Swiss National Science Foundation (to M.S., P2ZHP3_151580) and the Medical Research Council (to D.R.A.). We thank S. Suppmann, S. Uebel and E. Weyher-Stingl from the MPIB Biochemistry Core Facility and G. Sowa, S. Kroiss, K. Mayr and I. Paron from the department of Proteomics and Signal Transduction for providing technical assistance. M. Raeschle, F. Sacco, J. Liu and M. Murgia for critical reading and commenting on the manuscript. A. Itzen (TUM Munich) provided GroEL/ES chaperone plasmids. H. Cai (NIH Bethesda) provided LRRK2-/- mice. We thank T. Hochdorfer for protein purification as well as R. Gourlay and D. Campbell for technical support in the MRC-Protein Phosphorylation and Ubiquitylation Unit, the management of mouse colony service, genotyping and DNA Sequencing Service (coordinated by Nicholas Helps), the tissue culture team (coordinated by K. Airey and J. Stark) and the protein production and antibody purification teams (coordinated by H. McLauchlan and J. Hastie). Eli Lilly provided the LRRK2-G2019S$^{Lilly}$ knock-in mice. We acknowledge K. Basu, M. Miller and J. Scott from the chemistry department at Merck Research Laboratories and the Merck LRRK2 program team for their contributions leading to the discovery of MLI-2.

## Additional information

### Competing interests

SWi: Employee of GlaxoSmithKline, a global healthcare company that may conceivably benefit financially through this publication. MJF: Employee of Merck Research Laboratories. JAM: Employee of GlaxoSmithKline, a global healthcare company that may conceivably benefit financially through this publication. ADR: Employee of Merck Research Laboratories, Contributed unpublished essential data or reagents. The other authors declare that no competing interests exist.

### Funding

| Funder | Grant reference number | Author |
| --- | --- | --- |
| Schweizerischer Nationalfonds zur Förderung der Wissenschaftlichen Forschung | P2ZHP3_151580 | Martin Steger |
| Michael J. Fox Foundation for Parkinson's Research | 6986 | Matthias Trost<br>Dario R Alessi<br>Matthias Mann<br>Alastair D Reith |
| Max-Planck-Gesellschaft | | Esben Lorentzen<br>Matthias Mann |
| Medical Research Council | MC_UU_12016/3 | Dario R Alessi |

The funders had no role in study design, data collection and interpretation, or the decision to submit the work for publication.

### Author contributions

MS, Designed the experiments, Performed the experiments, Conducted all global MS experiments and bioinformatic analyses of MS data and discovered the regulated Rab phosphorylation, Conception and design, Acquisition of data, Analysis and interpretation of data, Drafting or revising the article; FT, GI, Designed the experiments, Performed the experiments, Discussed the results and

approved the manuscript, Acquisition of data, Analysis and interpretation of data; PD, Designed the experiments, Discussed the results and approved the manuscript, Analysis and interpretation of data, Drafting or revising the article; MT, Performed the experiments, Discussed the results and approved the manuscript, Analysis and interpretation of data; MV, SWa, Performed the experiments, Discussed the results and approved the manuscript, Acquisition of data, Analysis and interpretation of data; EL, Designed the experiments, Discussed the results and approved the manuscript, Conception and design, Analysis and interpretation of data; GD, SWi, Contributed essential reagents, Generated the G2019SGSK knock-in mouse line, Discussed the results and approved the manuscript, Contributed unpublished essential data or reagents; MASB, BKF, Generated A2016T knock-in mice and supported the project, Discussed the results and approved the manuscript, Conception and design, Contributed unpublished essential data or reagents; MJF, Employees of Merck Research Laboratories, Contributed unpublished essential data or reagents; JAM, Contributed essential reagents, Provided MLI-2, Discussed the results and approved the manuscript, Contributed unpublished essential data or reagents; ADR, Contributed essential reagents, Generated the G2019SGSK knock-in mouse line, Provided GSK2578215A and supported the project from conception, Discussed the results and approved the manuscript, Contributed unpublished essential data or reagents; DRA, Designed the experiments, Wrote the manuscript, Discussed the results and approved the manuscript, Conception and design, Analysis and interpretation of data, Drafting or revising the article; MM, Conception and design, Analysis and interpretation of data, Drafting or revising the article

### Ethics

Animal experimentation: All animal studies were ethically reviewed and carried out in accordance with Animals (Scientific Procedures) Act 1986, the GSK Policy on the Care, Welfare and Treatment of Animals, regulations set by the University of Dundee and the U.K. Home Office. Animal studies and breeding were approved by the University of Dundee ethical committee and performed under a U. K. Home Office project license.

## Additional files

### Supplementary files

• Supplementary file 1. All phosphosites quantified in the phosphoproteomic screens one (PS1, LRRK2-G2019S$^{GSK}$ MEFs and two (PS2, wt and LRRK2-A2016T MEFs).

• Supplementary file 2. All quantified proteins of wt and LRRK2-A2016T MEFs.

• Supplementary file 3. (A) Significantly modulated sites of PS1. (B) Significantly modulated sites of PS2.

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
