## [Decision Letter]

Thank you for submitting your work entitled "Phosphoproteomics reveals that Parkinson's disease kinase LRRK2 regulates Rab GTPases" for consideration by *eLife*. Your article has been reviewed by three peer reviewers, one of whom is a member of our Board of Reviewing Editors. The evaluation has been overseen by the Reviewing Editor and Ivan Dikic as the Senior Editor.

The reviewers have discussed the reviews with one another and the Reviewing Editor has drafted this decision to help you prepare a revised submission.

Summary:

Mutations in the LRRK2 kinase have been identified as main contributors to genetically caused Parkinson's disease. Despite extensive research on this kinase its physiological substrates remained elusive. In this paper, Steger and colleagues identify Rab GTPases (Rab10, Rab8, Rab3, Rab12) as new LRRK2 substrates. By making use of mouse and human cell lines, specific LRRK2 inhibitors, LRRK2 disease mutations and a powerful combination of Mass spectrometry and biochemistry, the authors are able to show that LRRK2 phosphorylates Rab GTPases in vitro and in vivo at their conserved switch 2 domain. This phosphorylation event decreases the binding affinity of Rabs towards GDIs and control their membrane insertion. Next, the authors examined whether phosphorylation of LRRK2 target RABs affect their binding properties. Thereto, the authors compared interactomes of RAB8A LRRK2-phosphomimicking and -ablating mutants and revealed preferential binding of a number of RAB8A-interacting proteins (e.g. the RAB8 GEF Rabin8 or the GDP dissociation inhibitors GDI1 and GDI2) bound to the phosphorylation deficient variant. The authors carried on to demonstrate that LRRK2-mediated phosphorylation of RAB8A reduced the GEF activity of Rabin8 towards RAB8A. Finally, the authors extended their finding on loss of GDI1/2 binding upon LRRK2 phosphorylation to RAB10 and RAB12. LRRK2-induced phosphorylation of T72-Rab8a inhibits rates of Rabin8-catalyzed GDP exchange 4-fold and decreases RAB8a-Rabin8 interaction. In addition, the authors used a panel of pathogenic LRRK2 mutants to demonstrate correlation between LRRK2-mediated RAB8A or RAB12 phosphorylation and GDI1/2 binding. PD-associated LRRK2 mutations would shift the membrane-cytosol balance of Rabs towards the membrane compartment, thereby causing accumulation of inactive Rabs in the membranes. All of the presented experiments are well designed, statistically analyzed and all of the important findings are backed up with extensive supporting material. Collectively, this work represents an elegant tour-de-force from proteomics phospho site mapping to mechanistic dissection of the functional consequence of LRRK2 target phosphorylation.

Essential revisions:

1) Throughout the text (starting with title and Abstract) the authors should rephrase statements generalizing Rab family members as LRRK2 substrates since only 4 out of 70 human Rabs were actually shown to be LRRK2 substrates in vitro and only for two of them the respective site has been detected in vivo in the presence of LRRK2 and/or RAB overexpression. Along these lines, the authors even show that there are substantial differences in the in vitro phosphorylation of RAB5B, RAB7A, RAB7L1 and RAB39 compared to RAB1B, TAB8A and RAB10. Are T72 of RAB1B and RAB8A also found phosphorylated in cells? In addition to the RAB10 and RAB12 sites shown in Figure 3, the authors should also provide the quantitative MS data for RAB8A pT72 and RAB1B p72. While the authors make quite some effort to identify LRRK2 substrates in a near-physiological context, the current undifferentiated wording about Rab GTPases being general LRRK2 substrates does not really fulfill this initial valid claim.

2) The authors should test if RAB8A and/or other RABs phosphorylation by LRRK2 on specific sites like T72 influences/disrupts its interaction with GDIs (similar to Figure 6) by using the orthogonal ribosome methods (rather than phosphomimetic).

3) The authors should confirm their model (Figure 7) by checking weather LRRK2 mediated RAB8 phosphorylation influences RAB8 cytosol-membrane shuttling, by cellular fractionation or immunofluorescent co-staining of RABs and membrane compartments.

---

## [Author Response]

*1) Throughout the text (starting with title and Abstract) the authors should rephrase statements generalizing Rab family members as LRRK2 substrates since only 4 out of 70 human Rabs were actually shown to be LRRK2 substrates in vitro and only for two of them the respective site has been detected in vivo in the presence of LRRK2 and/or RAB overexpression. Along these lines, the authors even show that there are substantial differences in the in vitro phosphorylation of RAB5B, RAB7A, RAB7L1 and RAB39 compared to RAB1B, TAB8A and RAB10. Are T72 of RAB1B and RAB8A also found phosphorylated in cells? In addition to the RAB10 and RAB12 sites shown in Figure 3, the authors should also provide the quantitative MS data for RAB8A pT72 and RAB1B p72. While the authors make quite some effort to identify LRRK2 substrates in a near-physiological context, the current undifferentiated wording about Rab GTPases being general LRRK2 substrates does not really fulfill this initial valid claim.*

We changed the title, Abstract and Introduction slightly to make it clear from the beginning that LRRK2 phosphorylates a small subset of Rabs and not all of them. Initially, we identify T73-Rab10 and S105-Rab12 as physiological LRRK2 phosphorylation sites (Figure 3). T72-Rab8 was not identified in our initial MS screen – probably because of its low abundance – but a more directed analysis revealed that T72-Rab8 is clearly a LRRK2 target. This is revealed by our experiments in three different systems (HEK293 GFP-LRRK2 flpIn cells, MEFs and mouse brain) where endogenously expressed Rab8 is subjected to phosphorylation by LRRK2 (Figure 3—figure supplement 2). In particular, we demonstrate (by immunoprecipitation of Rab8 from mouse embryonic fibroblasts and subsequent MS analysis) that endogenous levels of T72-Rab8 vary dependent on the activity of LRRK2. pT72-Rab8 levels are higher in G2019S cells as compared to wt (about three-fold) and even lower in LRRK2 knockout cells (Figure 3—figure supplement 2). Furthermore, by measuring the brain phosphoproteomes of LRRK2-G2019S mice, injected with either vehicle or MLI-2, we find that pT72-Rab8 is downregulated when LRRK2 is inhibited (Figure 3).

In contrast, while Rab1a/b are very efficiently phosphorylated on T75 by LRRK2 in vitro (Figure 2 and Figure 2—figure supplement 1), Rab1a is not phosphorylated in a LRRK2-dependent manner when expressed in HEK293 cells (depicted in the new Figure 3—figure supplement 2 in the revised manuscript). This indicates that in cells not all Rab GTPases with a conserved S/T residue in the switch II region are phosphorylated by LRRK2. We also changed the text accordingly (Results section under “A subset of Rabs are physiological LRRK2 substrates”)

*2) The authors should test if RAB8A and/or other RABs phosphorylation by LRRK2 on specific sites like T72 influences/disrupts its interaction with GDIs (similar to Figure 6) by using the orthogonal ribosome methods (rather than phosphomimetic).*

While the orthogonal ribosome method is an elegant and valuable tool for incorporating synthetic amino acids into recombinant proteins in bacteria, to our knowledge, this method is currently not well established for producing proteins containing phosphorylated threonines. Because both Rab8a and Rab10 are directly phosphorylated by LRRK2 on threonine sites, this would make the technology difficult to apply here, even if we had ready access to it.

We are aware that phosphomimetic S/T-E mutations can disrupt the protein structure and this could by itself lead to abrogation of protein binding. However, in the paper, we also show that increased phosphorylation of Rab8a by PD pathogenic LRRK2 variants trigger T72 hyperphosphorylation and, importantly, that this strongly diminishes its capacity to bind GDIs (Figure 6). Thus, the combination of phosphomimetic and activated kinase experiment, which both show the same outcome, makes it extremely unlikely that abrogation of binding is due to an artefactual structure change in the mutant protein.

*3) The authors should confirm their model (Figure 7) by checking weather LRRK2 mediated RAB8 phosphorylation influences RAB8 cytosol-membrane shuttling, by cellular fractionation or immunofluorescent co-staining of RABs and membrane compartments.*

We agree that this is indeed a very important point and thank the reviewers for the helpful suggestion. To prove that compromising the binding capacities of Rabs to GDIs causes a shift in the equilibrium between membrane-bound and cytosolic Rabs, we have now performed subcellular fractionation experiments. For this we mixed SILAC labeled cells expressing Rab8a-T72A or Rab8a-T72E in a 1:1 ratio before fractionation, minimizing variation due to sample preparation. After fractionation, we measured Rab8a protein abundances in both the cytosolic and the membrane fraction by mass spectrometry. This revealed that the pool of T72E mutant protein (which is compromised in GDI binding) is increased in the membrane fraction as compared to T72A. Importantly, at the same time, the fraction of T72E mutant in the cytosol was significantly reduced. Thus inhibiting the binding of GDIs to Rabs changes the membrane/cytosol distribution, directly validating our model. These results are shown in the new Figure 6 and described in the Results and Materials and methods.